# Influence of Wood Properties and Building Construction on Energy Demand, Thermal Comfort and Start-Up Lag Time of Radiant Floor Heating Systems

Álvaro Ruiz-Pardo [1], Enrique Ángel Rodríguez Jara [1], Marta Conde García [2] and José Antonio Tenorio Ríos [3,*]

1 Departamento de Máquinas y Motores Térmicos, University of Cadiz, Av. Universidad de Cádiz, 10, 11519 Cádiz, Spain; alvaro.ruiz@uca.es (Á.R.-P.); enrique.rodriguez@uca.es (E.Á.R.J.)
2 Departamento de Ingeniería Forestal, University of Córdoba, Ctra. Madrid, Km 396, 14071 Córdoba, Spain; marta.conde@uco.es
3 Eduardo Torroja Institute for Construction Sciences, Spanish National Research Council (CSIC), de Serrano Galvache, 4, 28033 Madrid, Spain
* Correspondence: tenorio@ietcc.csic.es

**Abstract:** Radiant floor heating is becoming increasingly popular in cold climates because it delivers higher comfort levels more efficiently than conventional systems. Wood is one of the surface coverings most frequently used in radiant flooring, despite the widely held belief that in terms of thermal performance it is no match for higher conductivity materials if a high energy performance is intended. Given that the highest admissible thermal resistance for flooring finishes or coverings is generally accepted to be 0.15 $m^2$K/W, wood would appear to be a scantly appropriate choice. Nonetheless, the evaluation of the thermal performance of wooden radiant floor heating systems in conjunction with the building in terms of energy demand, thermal comfort, and start-up period, has been insufficiently explored in research. This has led to the present knowledge gap around its potential to deliver lower energy consumption and higher thermal comfort than high-thermal-conductivity materials, depending on building characteristics. This article studies the thermal performance of wood radiant floors in terms of three parameters: energy demand, thermal comfort, and start-up lag time, analysing the effect of wood properties in conjunction with building construction on each. An experimentally validated radiant floor model was coupled to a simplified building thermal model to simulate the performance of 60 wood coverings and one reference granite covering in 216 urban dwellings differing in construction features. The average energy demand was observed to be lower in the wood than in the granite coverings in 25% of the dwellings simulated. Similarly, on average, wood lagged behind granite in thermal comfort by less than 1 h/day in 50% of the dwellings. The energy demand was minimised in a significant 18% and thermal comfort maximised in 14% of the simulations at the lowest thermal conductivity value. The vast majority of the wooden floors lengthened the start-up lag time relative to granite in only 30 min or less in all the dwellings. Wood flooring with the highest thermal resistance (even over the 0.15 $m^2$K/W cited in standard EN 1264-2) did not significantly affect the energy demand or thermal comfort. On average, wood flooring lowered energy demand by 6.4% and daily hours of thermal comfort by a mere 1.6% relative to granite coverings. The findings showed that wood-finished flooring may deliver comparable or, in some cases, higher thermal performance than high-conductivity material coverings, even when their thermal resistance is over 0.15 $m^2$K/W. The suggestion is that the aforementioned value, presently deemed the maximum admissible thermal resistance, may need to be revised.

**Keywords:** wooden radiant floor heating; radiant floor thermal modelling; energy efficiency of buildings; thermal comfort; natural stone vs. wood radiant floors

## 1. Introduction

The underfloor heating system is one of the oldest technologies for providing heating in buildings. The Ondol system was probably the first developed (Korea, 1000 BC) [1], and the Kang (China, 500 BC) [2] and the hypocaust (Greece/Rome 300 BC) [3] are other ancient technologies developed for this purpose. After the fall of the Roman Empire, underfloor heating disappeared from Europe during the Middle Ages, re-emerging in France and Britain in the 18th and 19th centuries [4]. In North America, the first radiant heat floor systems date from the end of the 19th century, but the popularization of these systems worldwide probably only began after the Second World War [5]. Radiant floor heating is currently one of the fastest growing heating systems, thanks in part to the introduction of plastic piping [6], systems based on heat pumps [7], and renewable energies.

Radiant floors have a number of advantages over other heating systems, including: high comfort levels [8], resulting from their ideal vertical temperature gradient [8,9]; high energy efficiency, due to the lower air temperature required to meet comfort conditions [9]; silent operation [8]; and their use of low water temperatures that are compatible with renewable energy [10–13]. For these reasons, radiant floor heating is gaining in popularity compared with more conventional systems [14,15].

Numerous investigations have been carried out on the thermal performance of radiant floors from different perspectives, such as control [13,16–18], thermal inertia [19,20], the use of PCMs [21–27], the arrangement of tubes [23,28], and the most suitable radiant surfaces [6], but fewer publications has been found on the influence of floor coverings properties, although, as Sattari and Farhanieh [29] showed in a parametric study, the thickness and thermal properties of the finish or covering have a major bearing on the performance of radiant floor heating.

The coverings most commonly applied in radiant flooring are porcelain or ceramic tiles, natural stone, or wood. Low-thermal-resistance natural stone and ceramic materials may initially be deemed to feature higher thermal performance than wood [30], which is nonetheless used as a covering in radiant floors for subjective reasons: its aesthetics tend to be more highly esteemed and it is deemed more pleasing to the touch. Another advantage, according to Zhao et al. [31], is that with wood 'the surface temperature is more moderate and uniform'. In an empirical study of four types of laminate wood flooring with conductivities ranging from 0.091 W/(m·K) to 0.12 W/(m·K), Seo et al. [32] found that when secured to the system with adhesives, wood delivered better results than when not secured. They contended that although more energy is needed for first-use start-up, such floors maintain a higher temperature for longer after the heating is turned off. That beneficial effect has also been studied in connection with renewable energies. Athienitis and Chen [33], for instance, studied the effect of the solar heat accumulating in wood floors.

European standard EN 1264-2, 'Water based surface embedded heating and cooling systems—Part 2: Floor heating: Prove methods for the determination of the thermal output using calculation and test methods', specifies that thermal resistance values in floor coverings ($R_{\lambda,B}$) exceeding 0.15 m$^2$K/W lie outside its scope, a contention reinforced in Part 3 (on dimensioning), according to which values of $R_{\lambda,B} > 0.15$ m$^2$K/W should be avoided. This has been interpreted to constitute a hard-and-fast maximum and to mean that coverings with thermal resistance higher than this value are unsuitable for radiant floors. Design engineers have consequently challenged the suitability of wood, with its fairly high thermal resistance, as the covering of choice for radiant flooring, particularly in the coldest climates.

In light of the above, although radiant floors with wood covering are widely used for applications other than energy, their use is ruled out when the primary aim is high energy performance. Nonetheless, the evaluation of the thermal performance of wooden radiant floor heating systems in conjunction with the building in terms of energy demand, thermal comfort, and start-up period, has been insufficiently explored in research. This has led to the present knowledge gap around its potential to deliver lower energy consumption and

higher thermal comfort than high-thermal-conductivity materials, depending on building characteristics.

This study assesses the impact of different wood coverings on the thermal performance of radiant floors compared to a high conductivity material in terms of three parameters: energy demand, thermal comfort and start-up lag time. The material chosen for comparison in this study was granite, insofar as it is characterised by one of the highest thermal conductivity values of all natural stones. The effect of covering thermal properties in conjunction with dwelling constructional characteristics on these parameters was explored.

Part 1 of the aforementioned European standard (EN 1264-1 [34]) on water-based, surface-embedded heating and cooling systems defines four radiant flooring layouts, which vary in terms of pipe position (Figure 1): embedded (types A and C), underfloor (type B), and heating floor with plane section system (type D). In type A, the pipes are embedded in the thermal diffusion layer, which also affords the system inertial and thermal diffusion. In type B, similar to A, the pipes are laid under the covering, where they rest on diffusor bands. Type C pipes are embedded in an underlayer crowned with a band made of a different material. While wood is explicitly listed as one of the possible covering materials for type B, its use with the other three types is not ruled out. The radiant flooring chosen for this study was standard-compliant for types A and C, configurations widely used in radiant flooring.

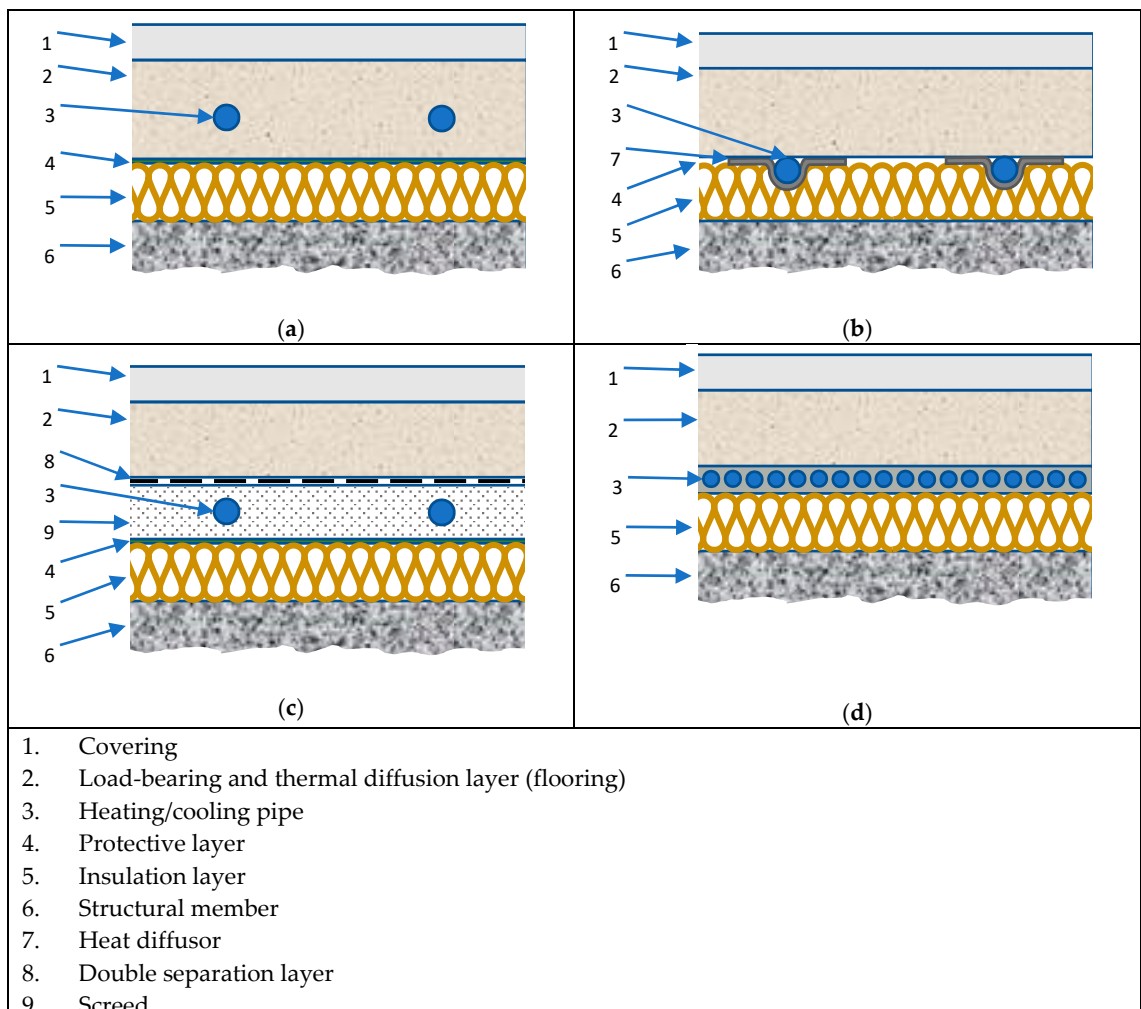

1. Covering
2. Load-bearing and thermal diffusion layer (flooring)
3. Heating/cooling pipe
4. Protective layer
5. Insulation layer
6. Structural member
7. Heat diffusor
8. Double separation layer
9. Screed

**Figure 1.** Types of radiant flooring by pipe layout (standard EN 1264-1). (**a**) Type A: Flooring with embedded pipes. (**b**) Type B: Flooring with underfloor pipes. (**c**) Type C: Flooring with embedded pipes. (**d**) Type D: Plane section systems.

In pursuit of the objectives of the study, a simplified building model was developed and coupled to a detailed, experimentally validated response-factor-method-based radiant floor model. A total of 60 simulations, one for each wood covering, was run with each of the 216 urban dwellings defined, and the results were compared to the performance simulated for a reference granite-finish flooring. Two types of simulation were performed: one to evaluate the energy demand and comfort, and the other to evaluate the start-up lag time. This combination of coverings, dwellings, and simulation types yielded a total of 26,352 simulations. The simulation models and the experimental validation of the radiant floor model are described in Section 2, while the simulation variables are discussed in Section 3, along with the radiant floor coverings, dwelling constructional features, and operating conditions. The findings are discussed in Section 4.

To broaden the applicability of the present findings, the case study was conducted for Madrid, a city comparable in terms of heating-degree days to the climate of a number of the world's major cities (Table 1).

**Table 1.** Cities with a number of heating-degree days at base T = 20 °C similar to Madrid's *.

|  | **Jan** | **Feb** | **Dec** |
|---|---|---|---|
| Chengdu (Ch) | 451 | 359 | 400 |
| Istanbul (TK) | 444 | 402 | 363 |
| London (UK) | 467 | 422 | 429 |
| Madrid (SP) | 452 | 364 | 434 |
| Tokyo (JP) | 423 | 367 | 352 |

* Source: author calculations using data drawn from Meteonorm [35], a commercial database.

## 2. Simulation Models

The models described in this section were developed to accurately calculate the thermal performance of radiant floor heating in a large number of cases in a reasonable time. The procedure followed was inspired by an idea put forward by Xu and Wang [36], in which a detailed model is applied for the element at issue (radiant flooring in this case) and a simplified model for the building housing it. The outcome was the development of two low computational-cost models, described in the paragraphs below. The detailed radiant floor model was based on the response factors method and the simplified building model on the lumped parameters or resistance capacity (RC network model).

### 2.1. Detailed Radiant Floor Thermal Modelling

Transient, two-dimensional modelling is required to simulate the thermal behaviour of embedded pipe structures [37]. The well-known response factor method [38] is applicable [39,40] for these purposes, since it delivers high calculation accuracy while ensuring processing speed by minimising the number of computational operations required.

The need to couple the heating to the building model and simulate control strategies entailed the use of a small time step in the simulations. A 15 min time step was defined for the response factor method used here to simulate the transient thermal response in radiant floorings. That the resulting radiant flooring model could be coupled to the simplified building model described in Section 2.2 with no need for iteration or matrix inversion also contributed to lowering computational costs.

The response factor method as initially proposed by Stephenson and Mitalas [41,42] is described in detail in [43]. In essence, it consists of obtaining the heat fluxes across three surfaces of a radiant floor (top, bottom, and pipe) over time in response to a unitary triangular temperature pulse on each surface, generating as output three sets of responses per surface excited. Such heat fluxes are known as response factors and must be determined analytically or numerically. Here, the widely used international ANSYS [44] software was applied to that end, performing the calculations with the finite element method, which is a detailed numerical method for solving the partial differential equations that govern transient two-dimensional heat conduction across radiant floors. ANSYS APDL derives

the surface distribution of heat flux on each surface of the radiant floor. It then internally calculates the averaged heat fluxes on each surface weighting the node values by the amount of element surface area associated with each node. These averaged heat fluxes are the response factors.

The response factors calculated were then used to calculate the actual heat fluxes in each surface of the radiant floor based on the superposition principle. Although a more detailed explanation is given in Appendix A, Equation (1), the general expression applied to calculate the heat flux across the three surfaces of the radiant floor is shown below:

$$\dot{q}_{s,k,floor}(n) = f_{1,k}(0)\, T_{s,1}(n) + f_{2,k}(0)\, T_{s,2}(n) + f_{3,k}(0)\, T_{s,3}(n) + h_k \tag{1}$$

where k denotes the top (1), bottom (2) or pipe (3) surface; n the simulation time step at issue; $f_{j,k}$ the response factor for surface k when a unitary triangular pulse is applied to surface j; and $h_k$ a term grouping the information on the preceding time steps for surface k.

### 2.2. Coupling the Building and Radiant Floor Thermal Models

The building simulations were conducted with a lumped parameter or resistance-capacitance (RC-network) model. 'RC models have proven to be advantageous and are widely used in modeling of thermal dynamics' [45] due to their low computational cost, their capacity to accommodate the main physical information on buildings, and better-than-acceptable accuracy, providing they are properly characterised [46]. This work aims to plausibly simulate the thermal performance of dwellings while using a radiant floor heating system. Vivian J. et al. [47] found that the deviation in the heating peak load of a first-order model (a lumped parameter model with one thermal capacitance) with respect to a detailed model falls between +8% and −6%, and the heating needs approximately +/−5%. They concluded that both first- and second-order models 'appear to reliably calculate the overall energy needs of buildings in both heating and cooling seasons'. On these grounds, the RC-network model presented here appears to be suitable for the purposes of this research. Any number of examples of the use of RC models for thermal simulations in buildings can be found in previous research (see [48–56]), as well as in studies that used such models in building simulations involving radiant flooring. Joe and Karava [57], for instance, developed a thermal resistance and capacitance model to optimise the control of radiant floor heating and Weber et al. [58] combined a detailed model for simulating a thermally activated building component system (TABS) with an RC model for the building to address an issue closely related to that discussed here. Thanks to the RC model applied to the building, which was readily coupled to the radiant floor model described in the preceding sub-section, the output for the 26,352 (Section 3.3) simulations conducted was obtained in a reasonable amount of time.

The simplified thermal model for the dwelling simulation coupled to the radiant floor model is illustrated in Figure 2. The bottom surface of the radiant flooring was assumed to be bounded by an indoor space forming part of the environment-controlled dwelling on the storey below. In the figure, $T_{s2}$ is the temperature of the bottom surface of the flooring and $h_{cr2}$ the convective–radiative heat transfer coefficient between that surface and the indoor space below at temperature $T_i$. The top surface, in turn, was assumed to be bounded by the apartment to be heated, consisting in outer walls and indoor partitions separating it from other dwellings. $T_{cr,I}$ represents the convective–radiative temperature of the space to be heated and hcr1 the convective–radiative heat transfer coefficient between the capacitance node of the space (node C) and the top surface of the flooring at temperature $T_{s1}$. $U_i$ represents the U-value of the indoor partitions between the apartment and the adjacent spaces at temperature $T_i$. $U_e$ represents the U-value of the outer walls, including windows, with the outdoor environment at linear air temperature, $T_{sa}$, which is defined as 'the outside air temperature which, in the absence of solar radiation, would give the same temperature distribution and rate of heat transfer through a wall (or roof) as exists due to the combined effects of the actual outdoor temperature distribution plus the incident solar

radiation' [59]. The internal loads and ventilation are represented as $\dot{Q}_{cr}$ at the capacitance node, which also included the thermal loads attributable to the solar radiation entering the space through the windows.

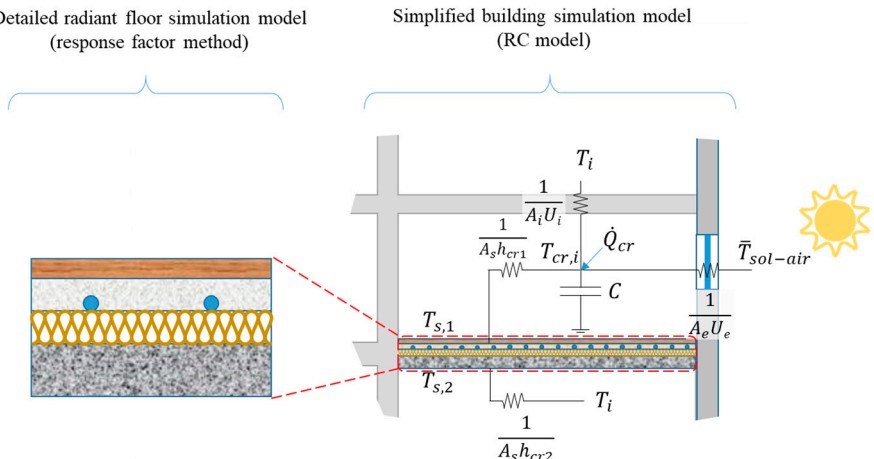

**Figure 2.** Radiant flooring model coupled to the building RC model.

The coupling between the radiant flooring and building model is described below. The boundary conditions in the flooring model were the top ($T_{s,1}$), bottom ($T_{s,2}$), and piping ($T_{s,3}$) surface temperatures and the responses to the heat fluxes across those surfaces ($q_{s,1}$, $q_{s,2}$ and $q_{s,3}$, respectively). The boundary conditions in the simplified building model were the indoor temperatures in the adjacent dwellings ($T_i$), the outdoor sol-air temperature ($T_{sol-air}$), the top and bottom flooring temperatures ($T_{s,1}$ and $T_{s,2}$, respectively), and the heat flux at the capacitance node ($\dot{Q}_{cr}$). The responses in this model were also the heat fluxes at the top and bottom surfaces of the flooring ($q_{s,1}$ and $q_{s,2}$, respectively), as well as the indoor temperature in the space to be heated, i.e., the temperature of the capacitance node ($T_{cr,i}$). As the figure shows, the two models were coupled via the heat fluxes at the top and bottom surfaces on the radiant flooring. The results of this coupling, i.e., the indoor environmental temperature $T_{cr,I}$ and the radiant floor surface temperatures $T_{s,1}$, $T_{s,2}$, and $T_{s,3}$ were found with no need for iteration or other numerical procedures. The values adopted for each pre-defined building parameter are given in Section 3.

The heat flux at the top surface of the flooring, $q_{s,1,build}$, was calculated with Equation (2):

$$\dot{q}_{s,1,\,build} = h_{cr,1}(T_{cr,i} - T_{s,1}) \tag{2}$$

for which the capacitance node temperature $T_{cr,I}$ needed to be found. The procedure to determine the temperature for the boundary conditions listed earlier is described in [60] and in Appendix A. The excitation was taken to vary linearly at every simulation time step, a reasonable premise inasmuch as excitation data are normally given as discrete values spaced at time-step intervals. Applying continuity across the top surface of the flooring, the conduction heat flux obtained with the flooring model (Equation (1)) is equal to the convective–radiative heat flux found with the RC building model (Equation (3)):

$$\dot{q}_{s,1,floor} = \dot{q}_{s,1,\,build} \tag{3}$$

$$f_{1,1}(0)\,T_{s,1} + f_{2,1}(0)\,T_{s,2} + f_{3,1}(0)\,T_{s,3} + h_1 = h_{cr,1}(T_{cr,i} - T_{s,1}) \tag{4}$$

Substituting the expression for temperature $T_{cr,i}$ obtained with the RC model for the building (Appendix A) and re-grouping terms in Equation (2) yielded the following simple algebraic equation:

$$a_1\,T_{s,1} + b_1\,T_{s,2} + c_1\,T_{s,3} + d_1 = 0 \tag{5}$$

where: $b_1 = f_{2,1}(0)$, $c_1 = f_{3,1}(0)$, $a_1 = f_{1,1}(0) + h_{cr,1}(1 - A_1)$ and $d_1 = h_1 - h_{cr,1} B_1$; A1 and B1 represent the known values at the current time step that depend on boundary conditions and the model parameters that define the building, i.e., heat capacity, C, and the thermal resistance between the space and the flooring surface. Explicit equations for A1 and B1, given in Appendix A, are omitted here for the sake of simplicity.

Similarly, applying continuity across the bottom and pipe surfaces yields:

$$f_{1,2}(0)\, T_{s,1} + f_{2,2}(0)\, T_{s,2} + f_{3,2}(0)\, T_{s,3} + h_2 = h_{cr,2}(T_i - T_{s,2}) \tag{6}$$

$$f_{1,3}(0)\, T_{s,1} + f_{2,3}(0)\, T_{s,2} + f_{3,3}(0)\, T_{s,3} + h_3 = h_{c,3}(T_w - T_{s,3}) \tag{7}$$

These equations can be re-grouped as shown below:

$$a_2\, T_{s,1} + b_2\, T_{s,2} + c_2\, T_{s,3} + d_2 = 0 \tag{8}$$

$$a_3\, T_{s,1} + b_3\, T_{s,2} + c_3\, T_{s,3} + d_3 = 0 \tag{9}$$

where $a_2 = f_{1,2}(0)$, $c_2 = f_{3,2}(0)$, $b_2 = f_{2,2}(0) + h_{cr,1}$ and $d_2 = h_2 - h_{cr,2}\, T_i$ in Equation (8) and $a_3 = f_{1,3}(0)$, $b_3 = f_{2,3}(0)$, $c_2 = f_{3,3}(0) + h_{c,3}$, and $d_3 = h_2 - h_{c,3}\, T_w$ in Equation (9).

Equations (4), (7) and (8) form a system of equations where the unknowns, $T_{s,1}$, $T_{s,2}$, and $T_{s,3}$, can be represented in a matrix as follows:

$$\begin{bmatrix} a_1 & b_1 & c_1 \\ a_2 & b_2 & c_2 \\ a_3 & b_3 & c_3 \end{bmatrix} \begin{bmatrix} T_{s,1} \\ T_{s,2} \\ T_{s,3} \end{bmatrix} = \begin{bmatrix} -d_1 \\ -d_2 \\ -d_3 \end{bmatrix} \tag{10}$$

The explicit expressions for temperatures $T_{s,1}$, $T_{s,2}$, and $T_{s,3}$, which depend on the response factors for the three flooring surfaces and the building's boundary conditions and parameters, can be readily found with Equation (10) by inverting the matrix. These flooring surface temperatures can then be used to calculate the temperature of indoor space $T_{cr,I}$ by Equation (A12).

As previously stated, the outputs of the model are the surface temperatures of the radiant floor $T_{s,1}$, $T_{s,2}$, and $T_{s,3}$, as well as the temperature of the indoor space $T_{cr,i}$. Another output of interest for the present work is the operative temperature Top. The operative temperature in a room is defined by standards CIBSE [61] and ISO 7730:2005 [62] as follows: 'the operative temperature in a real room is equal to the air temperature in an hypothetical room such that an occupant would experience the same net energy exchange with the surroundings'. These standards also mention that Top is used as an index temperature for comfort where air velocities are low, so it has been used in Section 4 for assessing the indoor thermal comfort. Top can be calculated according to ISO 7726:1998 [63] as:

$$T_{op} = \frac{\overline{T}_r \cdot h_r + T_{air} \cdot h_c}{h_r + h_c} \tag{11}$$

where $\overline{T}_r$ is the mean radiant temperature calculated as:

$$\overline{T}_r = T_{s,1} \cdot F_{s,1} + T_{walls} \cdot F_{walls} \tag{12}$$

In Equation (12) 'walls' refers to all the enclosures composing the space except the top surface of the floor. $F_{s,1}$ and $F_{walls}$ are the view factors of the occupant with floor top surface and space walls, respectively. As an approximation, $F_{s,1} = A_{s,1}/A_{total}$ and $F_{walls} = 1 - F_{s,1}$, where $A_{total}$ refers to the total area of the space enclosures, including the top surface of the floor. In the simplified building model indoor space is assumed to be at uniform temperature and $T_{walls} = T_{air} = T_{cr,i}$, so Top can be determined as:

$$T_{op} = \frac{T_{s,1} \cdot \left( \frac{A_{s,1}}{A_{total}} \right) \cdot h_r + T_{cr,i} \left[ \left( 1 - \frac{A_{s,1}}{A_{total}} \right) \cdot h_r + h_c \right]}{h_r + h_c} \tag{13}$$

### 2.3. Experimental Validation of the Radiant Floor Model

The radiant floor model developed was experimentally validated by using data from a laboratory trial conducted at Applus laboratories, in Barcelona, further to a request by Uponor Hispania, S.A.U., a specialist in radiant-floor-based HVAC systems. The trial, the details of which were furnished by Uponor, aimed, among other objectives, to experimentally measure the time taken to reach a comfortable temperature in a wood-covered radiant floor heating system. The temperatures recorded in the trials were used to validate the radiant floor model applied here.

The 90-millimetre-thick radiant floor heating system (Figure 3) comprised 19-millimetre-thick varnished oakwood planks secured with adhesives, a 30 mm anhydrite-based mortar slab with a 16 mm diameter embedded cross-linked polyethylene piping, and a smooth 25 mm enhancing-agent-modified EPS self-secured panel. The pipes were spaced at 150 mm in a 2 m × 2 m sample floor.

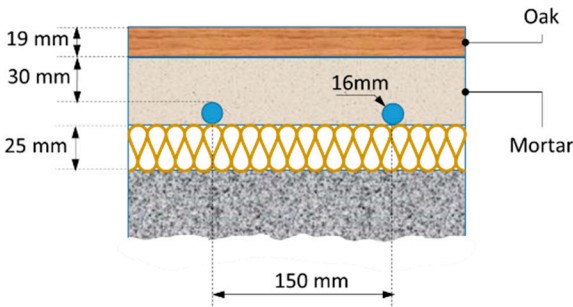

**Figure 3.** Cross-section of radiant flooring used in the model validation trials.

The trial was conducted in a 4.0 m × 4.0 m × 3.0 m chamber built to the specifications laid down in European standard EN 442-2 [64] for radiators and convectors, i.e., with five water-cooled sides and one insulated side fitted with the emitter.

Beginning at a temperature of 1 °C for both flooring and chamber, the trials consisted of pumping a given water flow, at a specified temperature, through the piping and determining the surface temperatures on the flooring and in the chamber until the latter reached 20 °C, measured at a reference point in the centre of the chamber 0.75 m above the flooring. Eight thermocouples were positioned on the wood covering to record the surface temperatures. The three stages of flooring construction depicted in the photographs in Figure 4 show the positions of the piping in the mortar and of the thermocouples on the wood covering. The trials were conducted at inflow water temperatures of 35 °C, 40 °C, and 45 °C, and a flow rate of 200 L/h, in line with European standard EN 1264 [64].

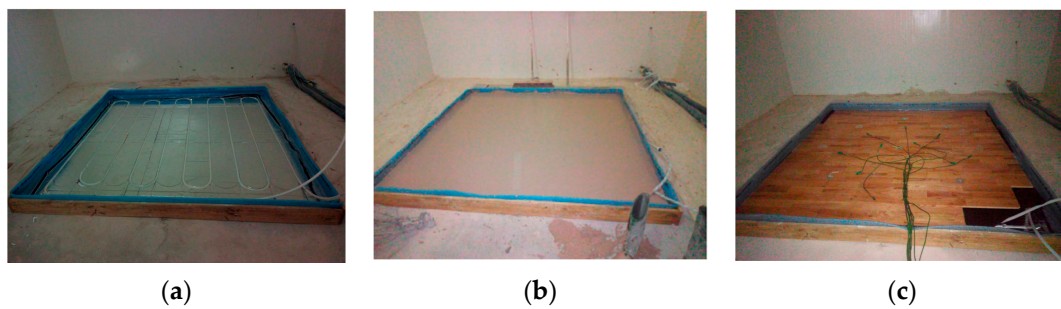

(**a**)            (**b**)            (**c**)

**Figure 4.** Trial flooring set-up. (**a**) Piping layout; (**b**) mortar layer; (**c**) floor covering and positions of surface-temperature sensors.

The accuracy of the transient thermal model for radiant flooring developed for this study was validated at inflow water temperatures of 35 °C, 40 °C, and 45 °C. The boundary conditions for each were defined by the parameters of inflowing water temperature, water

flow rate, and the experimentally measured air and wall temperatures of the chamber ($T_{cr}^{exp}$). The model output consisted of the flooring surface temperatures. For the purposes of validation, the flooring was thermally coupled to the chamber environment and the water. The water-side convective heat transfer coefficient was found with the Gnielinski correlation [65] for inner flow in smooth pipes. The convective–radiative heat transfer coefficient at the covering surface ($h_{cr}$) was found from the mean experimental covering temperature ($T_s^{exp}$) and the chamber temperature, along with the surface heat flux, ($\dot{q}_s$), calculated as laid down in the aforementioned standard EN 1264 [64]. For any given time, then, the convective–radiative coefficient was calculated as $h_{cr} = \dot{q}_s / \left( T_s^{exp} - T_{cr}^{exp} \right)$. For each inflow water temperature, the model-predicted surface temperature was compared to the mean surface temperature on the flooring recorded during the trials (Figure 5). The root mean square (RMS) error was then calculated between the model and experimental values for each water inflow temperature. The use of a single average temperature could have been a limitation of the model, since the tube arrangement in the experiment was serpentine, which could have led to the presentation of a less uniform temperature distribution on the surface compared to a spiral arrangement. However, during the experiments, the difference in water temperature between the inlet and the outlet was small (as an example, when the inlet temperature was 40 °C, the outlet was 39.4 °C, with small variations, during the test), so the temperature on the entire surface was very homogeneous. Therefore, the temperature distribution problem, for this case, was minimal. On these grounds, an average experimental soil temperature was obtained to compare with that of the model. Hence, the a priori limitation of the model, which only uses one temperature for the entire radiant floor surface, is irrelevant to its validation.

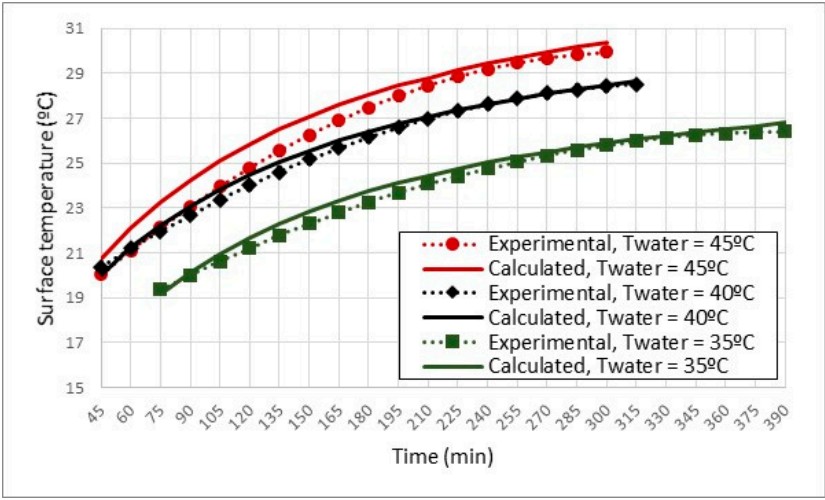

**Figure 5.** Experimental validation of the numerical model for radiant flooring.

The flooring surface temperatures found with the numerical model for water inflow temperatures of 35 °C and 40 °C fitted the experimental measurements very closely, with RMS errors of only 0.33 °C and 0.27 °C, respectively. The numerical and experimental values agreed less closely at a water inflow temperature of 45 °C, particularly in the early time steps, although the RMS error was only 0.75 °C. The less narrow fit for the inflow water temperature at 45 °C may be attributable to the use of an adiabatic boundary condition for the bottom side of the flooring. The downward losses in the flooring, which were greater for the inflow water at 45 °C than for the other values, were not be accounted for by the model. In light of the aforementioned findings, the accuracy of the numerical model was deemed to be valid for the present purposes.

## 3. Simulation Description

This section describes the radiant floor layout, the material properties and building construction designs envisaged in the simulations, and the two types of simulation conducted. The total number of simulations was the result of applying 61 (60 wood and 1 granite) coverings to 216 dwellings under two types of simulation or operating regime (61 × 216 × 2 = 26,352). As the mean time per simulation was around 5 s, the total calculating time amounted to about 37 computer hours (Hewlett-Packard, Intel® Xeon® processor; CPU E5-1620 V3 @ 3.50 GHz; 16.0 GB RAM).

### 3.1. Radiant Floor Layout and Material Properties

The flooring design applied in the simulations, depicted in Figure 6, was patterned on a typical radiant heating system that could be likened to layouts A and C in European standard 1264-1 [34], as noted earlier. The thermal properties of all except the covering (outer-most layer), which is described below, are listed in Table 2. The pipe wall thermal resistance was excluded from the simulations as negligible compared to the resistance of the surrounding mortar and insulation.

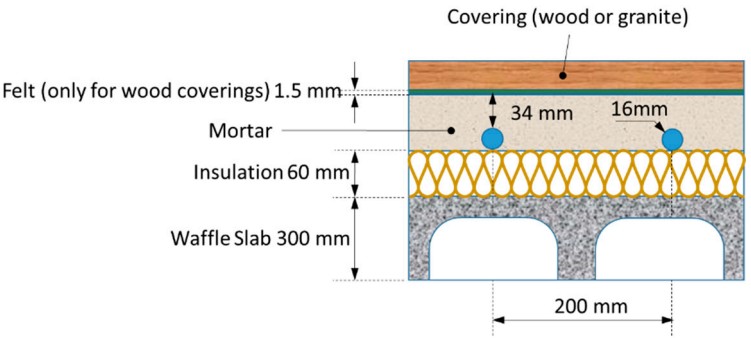

**Figure 6.** Radiant flooring layout.

**Table 2.** Thermal properties of materials comprising the radiant floor.

|  | Thermal Conductivity W/(m·K) | Density kg/m³ | Specific Heat J/(kgK) |
|---|---|---|---|
| Felt | 0.033 | 90 | 1000 |
| Mortar | 1.8 | 2100 | 2000 |
| Insulation | 0.033 | 30 | 1200 |
| Waffle slab | 1.22 | 1090 | 1000 |

The values for the thermal properties listed in Table 3 were drawn from an analysis of a number of commercial wood types used in flooring, specifically the types recommended for radiant floors by manufacturers such as Finsa, Pergo, Haro, Meister, and Junckers. Some of those manufacturers recommend a thermal resistance maximum of 0.15 m²K/W for coverings, whereas Spanish standard UNE 56810 on specifications for wood floors [66] sets the limit at 0.17 m²K/W.

The thermal resistance was consistently under 0.15 m²W/K in all the commercial floors analysed, while the thermal conductivity ranged from 0.10 W/m·K to 0.15 W/m·K for the thicknesses listed (Table 3).

Inasmuch as wood is a natural product, its characteristics vary, depending on its species and origin, the individual tree, or even the position of the wood in the trunk of a given tree. Thermal conductivity depends on several parameters, including microstructure, moisture and temperature, along with wood density [67,68], which is also related to most other physical and mechanical properties [69,70], including thermal conductivity.

**Table 3.** Commercial wood flooring thickness and thermal resistance values.

| Trade Name | Product | Thickness (mm) | Thermal Resistance (m²K/W) |
|---|---|---|---|
| Finsa. Finfloor | Laminate floor coverings | 8–10 | 0.06–0.154 |
| Pergo Lofoten–Senja–Langeland–Svalbard | Multi-layer parquet | 14 | 0.140 |
| Pero Laminate | Laminate floor coverings | 7–9.5 | 0.051–0.07 |
| Haro. Parquet 3000/3500/4000 | Multi-layer parquet | 11–13.5 | 0.063–0.110 |
| Haro. Tritty | Laminate floor coverings | 8 | 0.065 |
| Meister | Multi-layer parquet | 11–14 | 0.084–0.143 |
| Meister | Laminate floor coverings | 9 | 0.09 |
| Junckers | Solid hardwood planks | 15–20.5 | 0.09–0.12 |

The behaviour of possible wood coverings was analysed after compiling information from previous studies on density and thermal conductivity in temperate hardwood (oak, beech, ash, maple, walnut, cherry), softwood (pine, spruce, larch), and tropical wood (iroko, teak, jatoba, Merbau, awong). Denser natural species were not investigated.

On the grounds of the aforementioned inter-relationship between thermal conductivity and density, five mean densities were adopted based on the values found in previous studies (400 kg/m³, 500 kg/m³, 600 kg/m³, 700 kg/m³ and 850 kg/m³) with a view to covering a wide spectrum of wood properties. Three possible thermal conductivity values (consistent with earlier findings [14]) were assigned to each density, calculated from the mean conductivity values and standard deviations reported in previous studies [32,71–80], as well as from the commercial high thermal conductivity materials listed above. The findings are plotted in Figure 7, where the red dots denote the values adopted in this study.

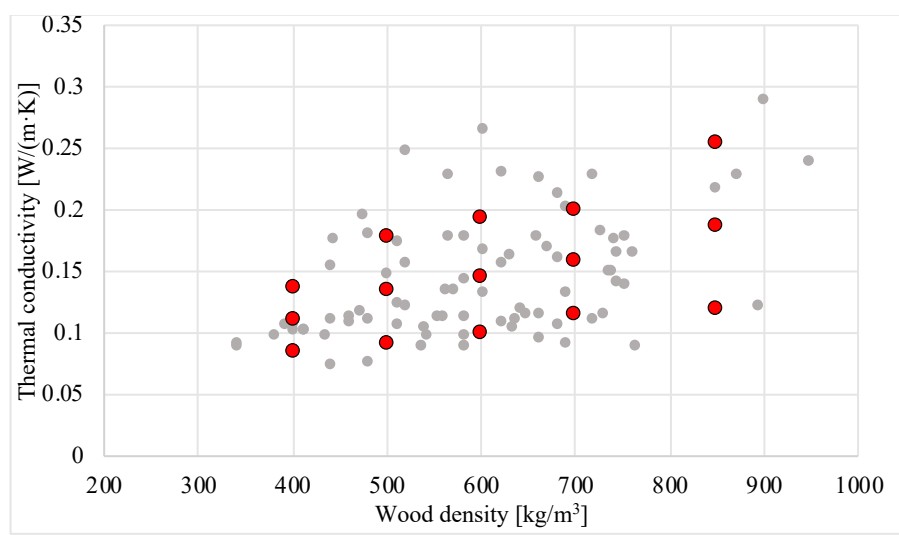

**Figure 7.** Wood properties: thermal conductivity and density (grey dots: values found in previous studies; red dots: values assumed here).

The fifteen thermal properties of the wood plotted in Figure 7 were each simulated for four thicknesses (10 mm, 15 mm, 19 mm, and 22 mm), yielding a total of 60 wood coverings.

A further simulation was run for a high-conductivity reference, a granite floor characterised by a thermal conductivity of 3.50 W/(m·K), a density of 2850 kg/m³, and a specific heat of 1.0 kJ/(kg·K), for comparison. This inclusion brought the total number of radiant floors simulated to 61.

### 3.2. Building Characteristics

The characteristics of the dwellings analysed are described in this section. The median area of urban dwellings in Madrid is 90 m$^2$ [81].

As a square enclosure measuring 9.5 m on each side, the dwelling simulated here had an area of 90.25 m$^2$, very close to the aforementioned median value. The storey height applied was 3 m. The dwellings were assumed to be located in one of two positions: at mid-building, with just one outer enclosure (hereafter 'interior dwelling'); or on the top storey, in a corner location (hereafter 'corner dwelling') (Figure 8).

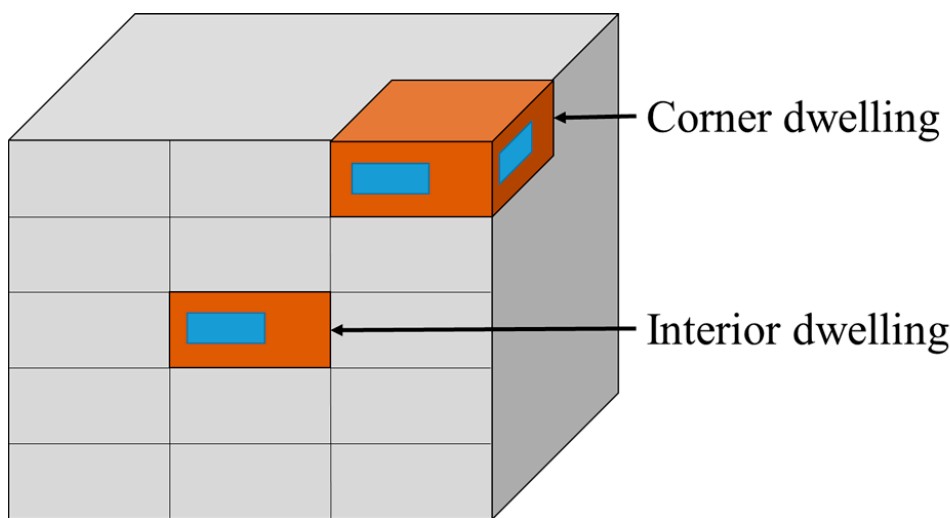

**Figure 8.** Dwelling locations used in simulations.

IIn pursuit of a wide variety of constructional characteristics, four variables were defined: glazed area, envelope insulation, overall heat transfer coefficient (U-value), and heat capacity and orientation, summarised below and substantiated in greater detail in Appendix B.

- Glazed area: 15%, 30%, or 80%;
- Envelope insulation (U-value): window, outer wall, and roof insulation were rated for the simulations as low, medium, or high depending on the respective U-values (Appendix B, Table A2), with high meaning better U-values than required by current legislation; medium meaning compliance-level; and low meaning non-conformity, i.e., as a rule, buildings 20 years old or over;
- Heat capacity: the three levels of heat capacity applied, low, medium, and high, were defined as per standard ISO 52016-1:2017 [82]);
- Orientation: the orientations adopted for interior dwellings were south, east, and west, and for corner dwellings, southeast, southwest, northeast, and northwest.

Consequently, the number of dwellings analysed was the result of combining the above parameters, namely two in-building locations, three percentages of glazing, three insulation levels, three levels of heat capacity, and four orientations, for a total of 216. The dwellings were coded as summarised in Figure 9.

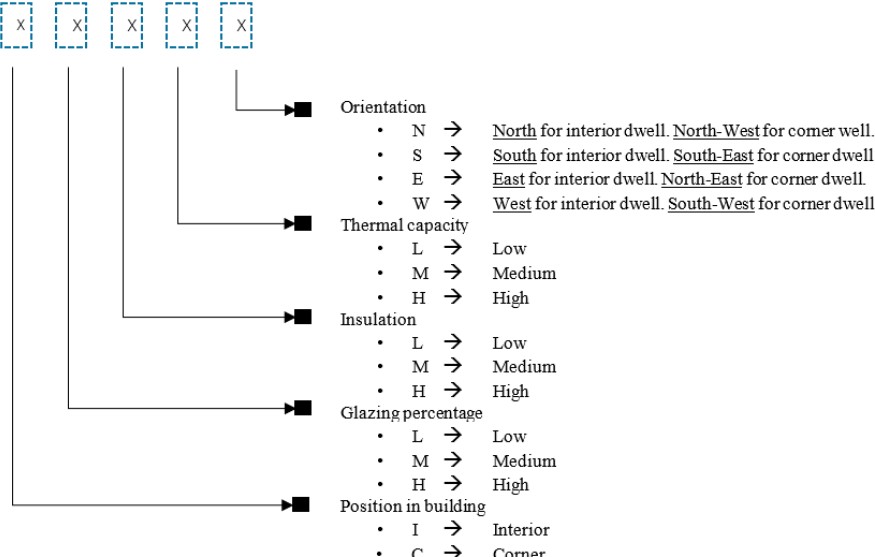

**Figure 9.** Dwelling labelling nomenclature.

*3.3. Simulation Types*

Two types of simulations were conducted, respectively designated 'normal regime' and 'start-up lag time'.

In normal-regime simulations, the radiant flooring was assumed to be operating in the conditions prevailing in January, the coldest month of the year. Under such conditions, the radiant flooring would be on from 08:00 to 23:00 to ensure the convective–radiative temperature, $T_{cr,I}$, required to maintain a set point temperature of 20 °C. The highest top surface temperature allowed by the control strategy was 29 °C, as specified in the standard ISO 1264-2:2009 [83].

In this simulation type, the model output primarily included the heat transferred by the water circulating in the pipes, used to assess the heating demand, and the indoor air and top-floor surface temperatures, to assess thermal comfort.

In the start-up lag time simulations, the radiant floor behaviour was simulated from the time it was switched on under given initial conditions until the 20 °C set point temperature was reached. The starting temperature for this simulation was determined by applying the conditions prevailing on the coldest day in January to a hypothetical three-day (72 h) period to envisage potential long lag times. The simulation ended after 72 h or when the set point temperature was reached, if earlier. To establish the initial conditions, January temperatures were assumed to vary freely and the dwelling to be subject to no occupancy loads. The aim of this simulation type was to calculate the time required to raise the room temperature to the set point from the initial free-float conditions.

## 4. Simulation Results and Discussion

The following discussion of the results is structured around the analysis of the three basic parameters used to assess performance: energy demand, thermal comfort, and start-up lag time.

Energy demand was determined as the amount of energy transferred by the water across the piping for the month of January. As thermal comfort is subjective [84] and cannot be unequivocally measured by any one parameter, a criterion had to be established on which to base the quantification. In this study, the criterion defined for this purpose was a minimum operative temperature of 20 °C, drawn from international standard ISO 7730 [62]. Based on the standard, spaces meeting this minimum wintertime requirement lie under comfort class B, with a PPD (predicted percentage dissatisfied) under 10%. Consequently, any operative temperature greater than or equal to 20 °C was assumed to be comfortable. The start-up lag time, in turn, was measured as the number of hours needed to raise

the indoor temperature to 20 °C from certain initial conditions previously determined by simulating dwellings in free-floating conditions for three consecutive days. The conditions determined from these free-floating simulations were then applied as the initial conditions for the start-up lag time simulations.

### 4.1. General Trends in Performance

The general trends observed in the three basic performance parameters are discussed in this section and plotted in Figure 10. The average, maximum, and minimum energy demand (Figure 10a), mean daily comfort hours (Figure 10b) and start-up lag time (Figure 10c) values were plotted for each dwelling by simulating all 60 woods, along with the value for the granite floor covering. In Figure 10a the dwellings are ranked along the *x*-axis from lowest to highest demand, in Figure 10b from highest to lowest number of comfort hours, and in Figure 10c from shortest to longest start-up lag time. Therefore, the dwellings are arranged in analogous but not identical order on the three figures, although, generally, the apartments with the lowest energy demand were observed to have a higher number of comfort hours and a shorter start-up lag time.

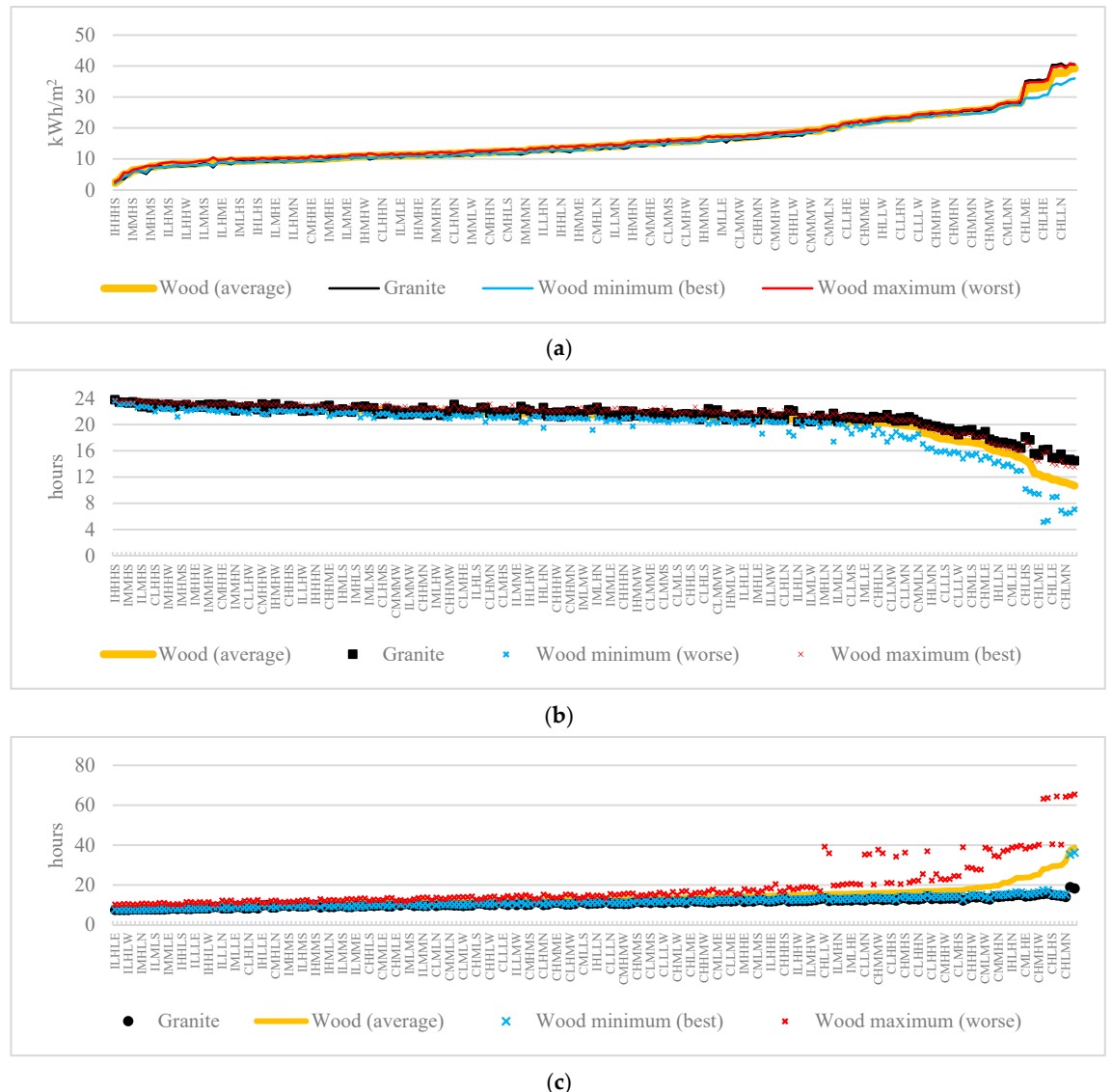

(a)

(b)

(c)

**Figure 10.** General trends of the basic performance parameters. (**a**) Energy demand (January); (**b**) thermal comfort: mean daily hours with $T_{op} \geq 20$ °C (January); (**c**) start-up lag time to reach $T_{op} = 20$ °C.

As expected, the dwellings with the best values for the three parameters (the dwellings placed furthest to the left in the figures) were predominantly those that were insulated the best and located at the interior, whilst the poorest performance (furthest to the right in the figures) was observed for the dwellings with the least insulation and located in corners. This pattern was observed in all the dwellings, whether with wood or granite floor coverings.

As the three figures show, by and large, the difference between using granite or wood covering was not significant, except in the most poorly insulated and corner-located dwellings. In many dwellings, at least one of the wood coverings exhibited lower demand or a greater number of comfort hours than the granite covering, whereas granite behaved consistently better in terms of start-up lag time. All the wood-floored dwellings with lower demand and more comfort hours than their granite-bearing counterparts were the best-insulated and located in the interior. On the whole, in corner-located and poorly insulated dwellings, granite out-performed wood in all three parameters, although in most cases, the values of at least one wood covering were observed to come close to those of the granite.

Specifically, the average energy demand was observed to be lower in the wood than in the granite coverings in 25% of the dwellings simulated. Similarly, on average, wood lagged behind granite in thermal comfort by less than 1 h in 50% of the dwellings. The average difference in start-up lag time between the wood and granite coverings was less than 3 h for 75% of the dwellings.

We found that, on average, with granite floors, the start-up lag time was just over 11 h, while with wooden floors it was 14 h, i.e., 27% longer. The better insulated the building, the lower these times and the difference between them were; in the best case, the time was 7.5 h for granite and 7.75 for wood (3.3% difference). The times were higher for buildings worse insulated, in the worst case 18 h for granite and 36 h for wood (112% difference). These data highlight the importance of the combined analysis of buildings and radiant floor heating. It is difficult to set a limit beyond which the start-up lag time is unacceptable, both in absolute terms, namely, the number of hours it takes to achieve the comfort condition, and in relative terms, namely, the difference between a high-conductivity floor such as granite and a wood floor. This is because the start-up time depends greatly on the specific use, e.g., if it is an occasional-use dwelling, such as a tourist flat, a start-up time of more than a couple of hours is probably unacceptable, but if it is a continuous-use flat, where the heating is usually on for most of the heating season, almost any start-up time can be acceptable.

### 4.2. Effect of Building Construction

The effect of building characteristics on the water-side energy demand (Figure 11) and thermal comfort delivered by the radiant floors (Figure 12) is addressed in this section. The construction parameters assessed here were dwelling heat capacity, insulation level, and percentage of glazed area, classified further according to the values listed in Appendix B. The impact of each parameter was assessed for dwelling orientation and location (corner or interior).

As shown in Figure 11, demand was observed to be greater in the corner-located dwellings. In the interior-located dwellings, whether wood- or granite-floored, demand was consistently highest among the dwellings facing north and lowest for those facing south. By contrast, demand varied very little between east- and west-facing interior-located dwellings. Demand declined and the difference between the average demands of wood and granite covering narrowed with rising building heat capacity, with the average demand exhibited by wood coverings in south-oriented dwellings varying by 37% from lowest to highest building heat capacity.

Demand also declined in all the interior-located dwellings with rising insulation levels. For wood coverings in south-oriented apartments, the average demand was 46% lower in the dwellings with the highest than in those with the lowest insulation; similar values were recorded for the other three orientations.

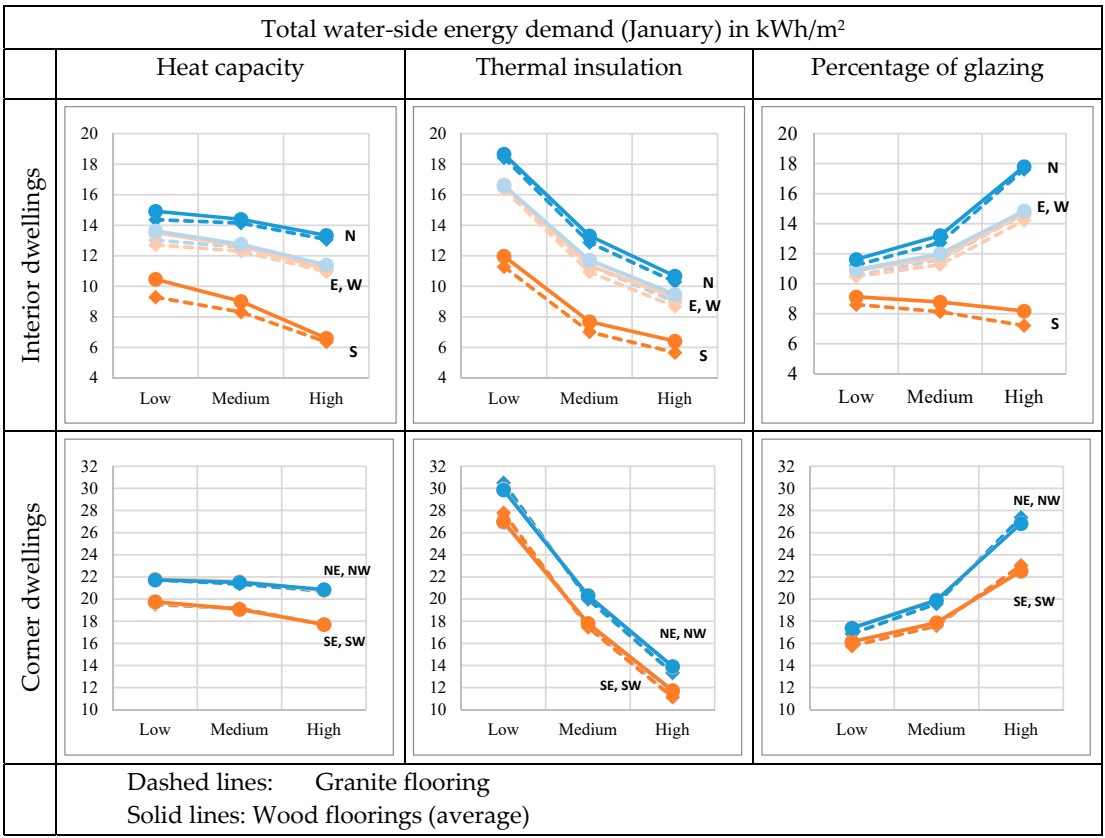

**Figure 11.** Effect of building construction on energy demand.

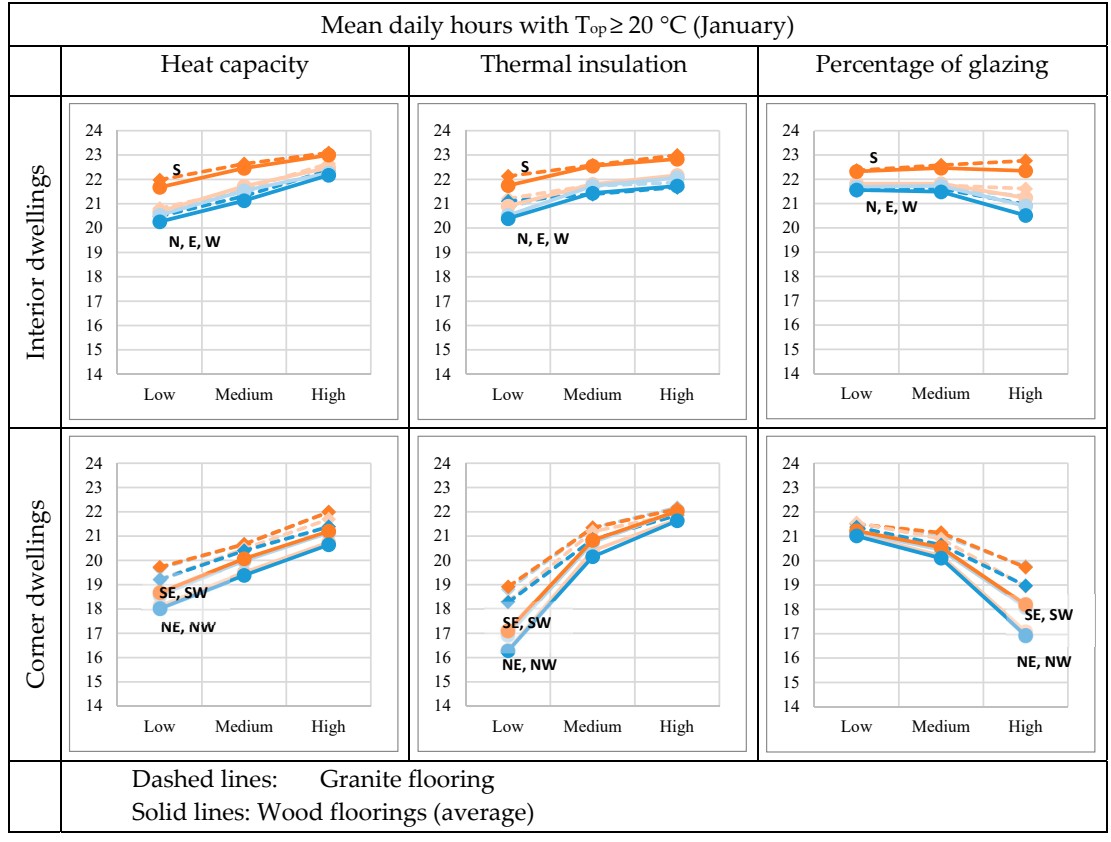

**Figure 12.** Effect of building construction on mean daily hours with Top ≥ 20 °C.

The effect of the percentage of glazing on energy demand exhibited a different pattern for south-facing interior dwellings than for the other three orientations. In the former, the solar gains afforded by the higher percentage of glazing offset the greater heat loss across the outer façade wall. In north-facing dwellings with wood coverings and the highest percentage of glazing, the average demand was 53% greater than with the lowest percentage of glazing, whereas in the most-glazed dwellings facing south, the average demand was 10% lower than in the buildings with the least glazing.

Corner-located dwellings facing northeast or northwest, whether wood- or granite-floored, exhibited higher demand than those facing southeast or southwest. Demand declined slightly with building heat capacity, with the average demand for wood coverings in southeast- and southwest-oriented dwellings varying by 10%, from lowest to highest capacity. Similarly, greater insulation levels lowered demand significantly more abruptly than heat capacity, with a 57% decline in average demand found for wood coverings from the lowest to the highest insulation level in southeast- and southwest-oriented dwellings.

The opposite pattern was observed for the percentage of glazing in all orientations: the higher the percentage, the higher the demand, even in south-oriented dwellings, where solar gains failed to offset the outward heat loss across façade walls. In northeast- and northwest-facing corner dwellings with wooden floor coverings, the average demand rose by 54% from the lowest to the highest percentage of glazing.

Based on the graphs in Figure 12, the interior-located dwellings exhibited more mean daily comfort hours than the corner-located dwellings. In the former, whether the flooring was covered with wood or granite, the comfort hours were higher in the south-facing dwellings than in those oriented in any other direction. The higher the heat capacity, the greater the number of comfort hours: in wood-floored, north-oriented dwellings, space conditions were comfortable for around 2 more hours when heat capacity was highest than when it was lowest.

The number of hours also rose consistently with rising insulation level. In north-oriented dwellings with wood floor coverings, space conditions were comfortable approximately 2 more hours in the most than in the least insulated.

In general, the mean number of daily comfort hours was impacted less significantly by the percentage of glazing than by the heat capacity or insulation level. In south-facing dwellings, this number was the same, irrespective of the percentage of glazing, although in wood-covered dwellings facing north, the difference between the lowest and highest percentage was 1 h.

The mean number of daily comfort hours was greatest in corner-located dwellings facing southeast and southwest for wood and granite coverings alike. In such dwellings, a higher heat capacity was also associated with more comfort hours. High-capacity corner dwellings with wood finish floorings had comfortable space conditions for 2.5 h longer than low-capacity dwellings, irrespective of orientation.

The number of comfort hours also rose consistently with rising insulation levels. In all the wood-floored corner dwellings, irrespective of orientation, space conditions were comfortable for 5 h longer when insulation was strongest than when it was weakest.

A higher percentage of glazing in corner locations resulted in a smaller number of hours of comfort in all orientations. For wood coverings in dwellings facing northeast and northwest, the number of comfort hours declined by 4 h between the lowest and highest percentages of glazing.

*4.3. Effect of Wood Thermal Conductivity*

The effects of the wood thermal properties on energy demand, thermal comfort, and start-up lag time are addressed in this section. The conductivity values that performed best on these three parameters are shown.

The thermal inertia (density, specific heat, and conductivity) of all the materials was envisaged in the simulations. Nonetheless, this section addresses the effect of the covering thermal conductivity only in light of the relationship between wood thermal conductivity

and density (Section 3.1) and the fact that the same specific heat was considered in all the different wood types. Consequently, variations in thermal conductivity encompass the effect of thermal inertia.

Unfortunately, no simple answers were found to the question of the thermal conductivity values that minimised energy demand or maximised the mean daily number of comfort hours. This lack of cut-and-dried findings, attributable to the difference in the roles played by the dwelling construction characteristics and covering thickness and properties depending on the case, ruled out any straightforward establishment of the conductivities consistently delivering minimum energy demand or maximum comfort.

Figure 13 plots the percentage of cases in which a certain range of thermal conductivities provided the lowest energy demand or the greatest thermal comfort. The bars in Figure 13a denote the percentage of cases in which the lowest energy demand co-existed with operative temperatures of more than or equal to 20 °C for at least 14 h per day. The aim was to find out the conductivity values that reduced energy demand in dwellings where comfort conditions were maintained. Figure 13b, in turn, plots the percentage of cases reaching the highest mean daily number of hours with an operative temperature of higher than or equal to 20 °C. According to the findings in Figure 13a, energy demand was minimised primarily when thermal conductivity was higher than 0.20 W/(m·K), although, in a significant 18% of cases, demand was minimised at conductivities of under 0.1 W/(m·K). The data in Figure 13b show that when wood conductivity was highest (0.20 W/(m·K) to 0.26 W/(m·K)), comfort was maximised in a greater percentage of cases than when it was lowest, although in this case, the highest comfort levels were found in 14% of cases with conductivities under 0.10 W/(m·K) and 16% of cases with values of 0.10 to 0.12 W/(m·K). Therefore, as wood with low thermal conductivity does not necessarily entail poor thermal performance, its use does not need to be ruled out. Rather, depending on the case, it could even be one of the highest performers.

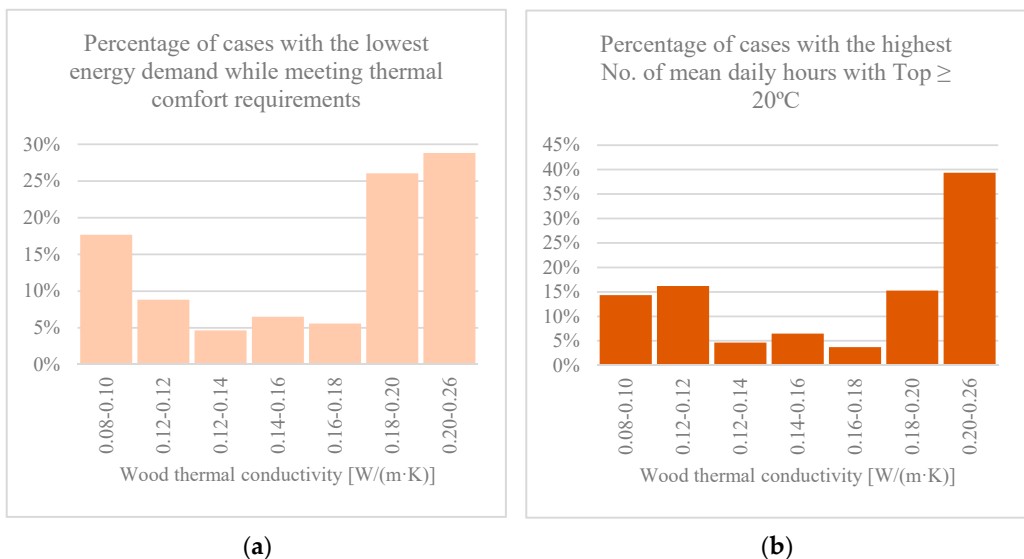

**Figure 13.** Percentage of cases in which a certain range of thermal conductivities performed best in terms of: (**a**) energy demand and (**b**) mean daily comfort hours.

The behaviour observed in Figure 13a is associated with water-side thermal power and the amount of time that the radiant floor heating must be switched on. In some cases, the demand was lower in low-conductivity coverings because they operated at lower power for longer times than high-thermal-conductivity finishes, which operated at higher power for fewer hours. The suggestion is that the use of high-conductivity wood is not always necessary to comply with thermal comfort criteria at low levels of energy demand.

In many cases, low-thermal-conductivity wood meets these criteria while minimising energy demand.

The curves in Figure 14 plot the effect of wood thermal conductivity on start-up lag time for all the dwellings analysed. Line 1 in the figure depicts the conductivity values minimising the start-up lag time; line 2 those prompting a delay of no more than 15 min over the line 1 time (15 min criterion); and line 3 those retarded by no more than 30 min relative to the line 1 time (30 min criterion). The covering consistently observed to minimise start-up lag time (line 1) to comfort conditions in all dwellings was the one with the highest thermal conductivity (0.26 (W/(m·K)). The explanation is simple: raising the temperature in a space in a short period of time depends on the power transferred from the water pipes across the floor covering. Nonetheless, in some dwellings, woods with lower conductivities delivered comfort at lag times that were not substantially longer (lines 2 and 3) and could therefore be adopted as the material of choice. Conductivities of 0.18 W/(m·K) often sufficed to meet the 15-min criterion in well-insulated, low-energy-demand dwellings, and in nearly all cases, such a conductivity value lay within the 30-min criterion. In a few well-insulated, interior-located dwellings, a conductivity value of 0.14 W/(m·K) sufficed to comply with the 30-min criterion.

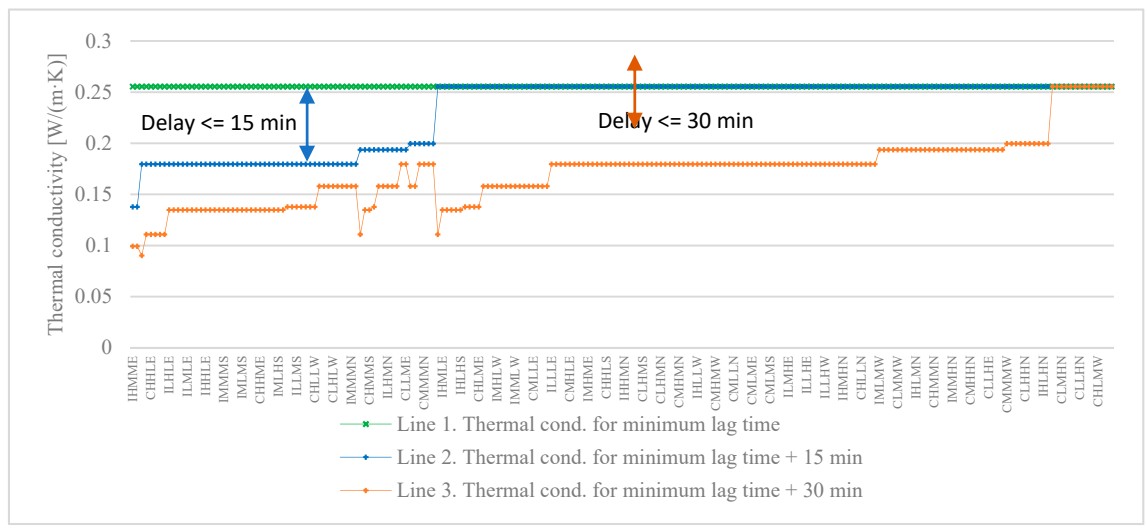

**Figure 14.** Thermal conductivities minimising start-up lag times.

As a rule, conductivity values under 0.26 W/(m·K) complied with the 15-min criterion in well-insulated, interior-located low-thermal-inertia dwellings (the furthest left in the figure), and with the 30-min criterion in nearly all the dwellings.

### 4.4. Effect of Wood Thermal Resistance

Given that the thermal resistance in wood coverings of radiant floors is a value routinely used to determine whether these coverings meet the minimum requirements that are thought to ensure correct system operation, the effect of that value on radiant floor performance was studied here.

Floor covering thermal resistance is plotted against energy demand in Figure 15a and thermal comfort in Figure 15b. The points on the three curves in Figure 15a represent, in series 1, the one dwelling of all those simulated that, at the appropriate thermal resistance, exhibited the highest demand; in series 2, these points represent the lowest demand. The series 3 curve denotes the average demand for all the dwellings. Figure 15b plots the mean daily hours during which the operative temperature was higher than or equal to 20 °C for the dwellings depicted in Figure 15a, i.e., each point in a given series in Figure 15b represents the same dwelling in the analogous series in Figure 15a. The hollow blue dots in series 1 denote non-compliance, i.e., cases where the operative temperature was greater than or equal to 20 °C for fewer than 14 h. The lowest thermal resistance value was recorded

for the granite floor coverings, while all the other dots plot the findings for the dwellings with wood coverings.

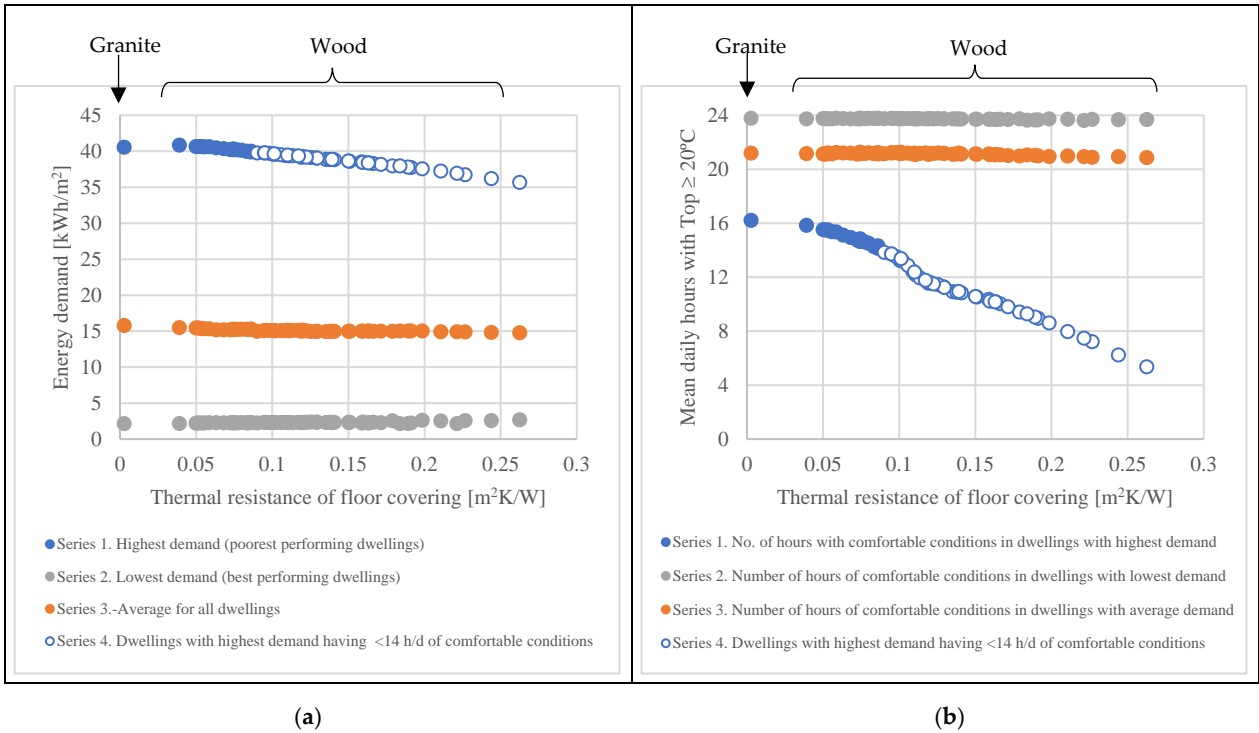

(**a**)　　　　　　　　　　　　　　　　　　(**b**)

**Figure 15.** Effect of covering thermal resistance on (**a**) energy demand and (**b**) mean daily comfort hours in the dwellings plotted in (**a**).

According to Figure 15a, the thermal resistance in the covering had no significant effect on energy demand in the series 2, low-energy-demand (often well-insulated and interior-located) dwellings, or on average energy demand (series 3). By contrast, demand declined with rising thermal resistance in the series 1, high-energy-demand dwellings. The explanation is to be found in the lower amount of energy transferred from the water pipes to the indoor space at higher thermal resistance levels. This decline in heat transfer also translated into a smaller number of comfort hours (series 1, Figure 15b). In these high-energy-demand dwellings, the number of mean daily comfort hours declined with rising resistance and dipped below the minimum 14-h requirement at values of over 0.09 $m^2$K/W. Conversely, in low-demand dwellings (series 2), comfortable conditions prevailed for nearly 24 h daily, irrespective of the thermal resistance of the wood covering, while the average value (series 3) observed was 21 h, also regardless of the thermal resistance values. On average, wood flooring lowered energy demand by 6.4% and daily hours of thermal comfort by a mere 1.6% relative to granite coverings.

As a rule, demand and comfort are inter-related: in buildings with generally low demand, comfort levels are high, and vice-versa. Depending on the dwelling, wood-covering thermal resistance was observed to make practically no difference to the energy demand or mean daily hours of thermal comfort, exhibiting the same, or very nearly the same numbers as granite floors. The conclusion drawn is that a thermal resistance of more than 0.15 $m^2$K/W in the covering (established as a maximum standard value) is compatible with efficient radiant floor heating.

The effect of wood-covering thermal resistance on start-up lag time is plotted in Figure 16. In this figure, the points on the series 1 curve represent a given dwelling that, at the appropriate thermal resistance, takes the longest, in number of hours, to raise the operative temperature to 20°; the points on the series 2 curve represent the dwellings taking the least time at each thermal resistance. Series 3 plots the average number of hours

required to raise all the dwellings to the operative temperature at a given resistance. Again, the granite flooring displayed the lowest thermal resistance value. Thermal resistance was observed to have a non-negligible effect in the dwellings with the shortest start-up lag times (series 2), rising by 37% from the 7.5 h recorded for the granite coverings to 10.25 h for the wood coverings with the highest thermal resistance. The effect on the average start-up lag time (series 3) was also significant, due to the substantial 64% longer lag time in the highest thermal resistance wood cover than in the granite cover. Thermal resistance also induced a steep rise in this parameter in the series 1 dwellings, those with longest start-up lag times, a pattern explained in Appendix C. At resistance values of over 0.14 m²K/W, comfortable conditions were not reached in these dwellings within the 72 h limit established in this study. Wood thermal resistance had a much more significant impact on start-up lag time than it had on energy demand and comfort hours.

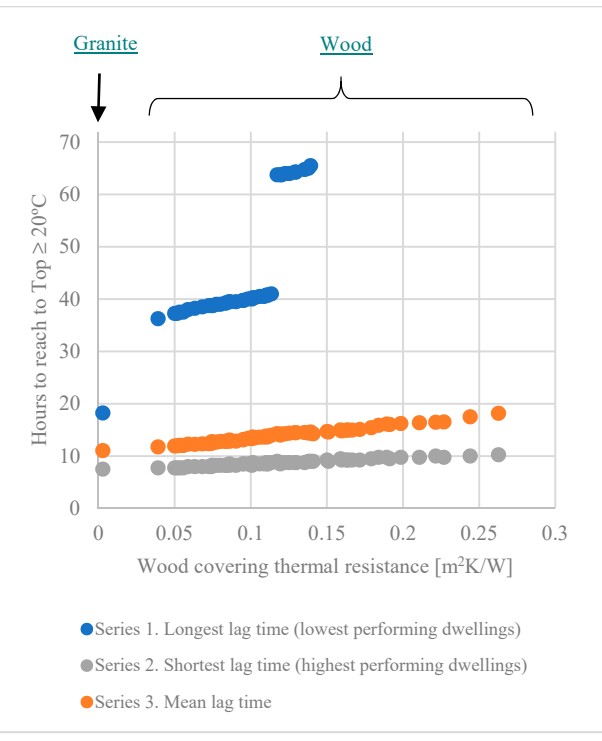

**Figure 16.** Effect of covering thermal resistance on start-up lag time.

## 5. Conclusions

Earlier studies on the most prominent variables involved in radiant flooring design established covering properties and thickness as the primary determinants of thermal performance.

The coverings most commonly laid over radiant flooring are porcelain or ceramic tiles, natural stone, and wood. Given their high thermal conductivity, natural stone and ceramic materials may initially be expected to be more thermally favourable than wood. Design engineers have consequently challenged the suitability of wood, with its fairly high thermal resistance, as the covering of choice in radiant floors, particularly in the coldest climates.

This study assessed the impact of different wood coverings on the thermal performance of radiant floors compared to high-conductivity natural stone (granite) in terms of three parameters: energy demand, thermal comfort, and start-up lag time. The effect of the coverings' thermal properties, in conjunction with the building construction characteristics, on these parameters was explored.

A simplified building model was developed and coupled to an experimentally validated radiant flooring model and used to run a total of 26,352 simulations. The total simulation computing time came to 37 h.

The main conclusions that may be drawn from the results are as follows:

- For most dwellings, the thermal properties of the wood affected energy demand and thermal comfort only scantly. Wood coverings delivered mostly similar and, in some cases, better results than granite coverings in those two respects. The impact of wood properties on demand and comfort was only significant for corner-located, poorly insulated dwellings. In such cases, granite flooring exhibited consistently higher thermal performance, although when appropriate wood properties were chosen, they proved to be a very close competitor to granite. These findings were not always associated with high thermal conductivity only. The average energy demand was observed to be lower in the wood than in the granite coverings in 25% of the dwellings simulated. Similarly, on average, wood lagged behind granite in thermal comfort by less than 1 h in 50% of the dwellings.

- Wood properties played a more substantial role in start-up lag times than in demand or thermal comfort, although the general pattern was much the same: for most dwellings, none of the wood radiant floors simulated lengthened the lag time substantially. As a rule, the dwellings where energy loss was greatest (corner dwellings, those with medium or high percentages of glazing, those that were minimally or only moderately insulated and oriented toward the north, east, or west) required a suitable choice of wood properties to prevent lag times from rising inordinately. The average difference in start-up lag time between the wood and granite coverings was less than 3 h for 75% of the dwellings. It is not possible to set a general time limit beyond which the start-up lag time is unacceptable, as this depends on the use of the dwelling, e.g., whether it is for tourism or for continuous use.

- Despite the scant impact of wood properties in most cases, the pursuit of simple rules to determine which properties would be the most suitable under given circumstances proved to be futile because the combination of wood properties, thickness, and dwelling construction characteristics followed no consistent pattern. The conclusion drawn, therefore, was that cover properties should be studied case-by-case to determine those expected to deliver the best thermal performance.

- In most cases, the highest thermal conductivity values were found to minimise energy demand, maximise comfort, and shorten start-up lag times. Energy demand was minimised primarily when thermal conductivity was higher than 0.20 W/(m·K), although in a significant 18% of cases, demand was minimised at conductivities of under 0.1 W/(m·K). When wood conductivity was highest (0.20 W/(m·K) to 0.26 W/(m·K)), comfort was maximised in a greater percentage of cases than when it was lowest, although in this case, the highest comfort levels were found in 14% of cases with conductivities under 0.10 W/(m·K) and 16% of cases with values of 0.10 to 0.12 W/(m·K). Conductivities of 0.18 W/(m·K) often increased the start-up lag time by only 15 min in well-insulated, low-energy-demand dwellings and in nearly all such dwellings, the conductivity value increases it by 30 min. On these grounds, wood with high thermal conductivity cannot be said to always be necessary for the design of radiant floors.

- One of the conclusions of this study that may be of most immediate interest is that the lowest thermal conductivity and thickest floor covering, i.e., wood flooring with the highest thermal resistance (even more of 0.15 m²K/W value) does not significantly affect the energy demand or thermal comfort. On average, wood flooring lowered energy demand by 6.4% and daily hours of thermal comfort by a mere 1.6% relative to granite coverings.

- The findings on the thermal resistance of wood coverings provided no justification for establishing an upper limit that must not be exceeded in the selection of woods for radiant floors. Although European standard EN 1264-2 [85] makes no provision for coverings with thermal resistance values of over 0.15 m²K/W, they are not explicitly prohibited. In fact, thermal resistance values higher than 0.15 m²K/W did not raise energy demand significantly, nor did they lower the number of comfort hours in the vast majority of the conditions simulated. This study consequently suggests that the

standard should be revised and the reference to that value deleted, since manufacturers have misconstrued it to be a limit not to be exceeded in the design of wood-covered radiant flooring.

- It was shown that the thermal behavior of radiant floor heating systems is closely linked to building conditions and, therefore, it is necessary to carry out a technical study for each particular case.

Our approach is based on the following three variables: (1) energy demand, (2) thermal comfort, and (3) start-up lag period. From our point of view, with these three variables, the overall performance of radiant floor heating can be clearly evaluated when its behaviour in transient mode coupled to the building needs to be assessed. The objective, as with any heating system, is to achieve thermal comfort for as long as possible with the lowest energy consumption. This can be analysed by using these two variables: the time in which comfort conditions are achieved and the energy demand of the radiant floor. On the other hand, the time required to achieve thermal comfort from an initial condition that is far from comfortable can be relevant for certain applications. Other approaches, such as the analysis of the power of the systems, or the surface temperature distribution of the floor (for hot and cold spots) may be of interest for other specific studies.

Further work along these lines should be conducted, since the present case study yielded results applicable only to the climate of Madrid and similar locations. The research should be extended to address other locations to determine whether the conclusions are climate-dependent.

**Author Contributions:** Á.R.-P.: methodology, software, investigation, data curation, formal analysis, writing—original draft. E.Á.R.J.: methodology, software and validation, investigation, data curation, formal analysis, writing—review and editing. M.C.G.: investigation, validation, writing—review and editing. J.A.T.R.: conceptualization, methodology, validation, formal analysis, supervision, writing—review and editing. All authors have read and agreed to the published version of the manuscript.

**Funding:** This research was funded by the European Agricultural Fund for Rural Development (EAFRD) and Spain's Ministry of Agriculture, Fishing and Food under the project 'Grupo Operativo Madera Construcción Sostenible', ref. 20180020012335.

**Institutional Review Board Statement:** Not applicable.

**Informed Consent Statement:** Not applicable.

**Acknowledgments:** The information for validating the model furnished by radiant flooring industry leader Uponor is gratefully acknowledged.

**Conflicts of Interest:** On behalf of all the authors, the corresponding author states that there is no conflict of interest.

## Nomenclature

| | | |
|---|---|---|
| $A$ | Area | m$^2$ |
| $ACH$ | Air changes per hour | h$^{-1}$ |
| $C$ | Thermal capacitance | J/K |
| $C_P$ | Specific heat | J/(kgK) |
| $f$ | Response factors of the radiant floor | |
| F | View factor | |
| $GR$ | Solar incident global irradiation | W/m$^2$ |
| $h$ | Convective heat transfer coefficient | W/(m$^2$K) |
| $\dot{q}$ | Heat flux density | W/m$^2$ |
| $\dot{Q}$ | Heat transfer rate | W |
| $R$ | Thermal resistance | (m$^2$K)/W |
| $SHGC$ | Solar heat gain coefficient of windows, including frames | |
| $t$ | Time | s |
| $T$ | Temperature | °C |

| | | |
|---|---|---|
| $U$ | Overall heat transfer coefficient | W/(m$^2$K) |
| $\dot{V}$ | Volumetric air flow rate | m$^3$/s |
| **Greek symbols** | | |
| $\rho$ | Density | kg/m$^3$ |
| **Subscripts** | | |
| air | Air | |
| build | Building | |
| c | Convective | |
| cr | Convective and radiant | |
| eq | Equivalent | |
| e | Exterior enclosures | |
| floor | Floor | |
| i | Indoor, Interior partitions | |
| r | radiant | |
| s | Surface | |
| sa | Sol-air | |
| w | Window | |
| walls | walls | |
| 1 | Top surface of radiant floor | |
| 2 | Bottom surface of radiant floor | |
| 3 | Pipe surface of radiant floor | |

## Appendix A. Development of the Simulation Models

This appendix describes in more detail the simulation models used in this study.

### Appendix A.1. Detailed Radiant Floor Thermal Model

As described in Section 2.1, radiant floor heating was modelled using the response factor method to calculate the heat fluxes across the three surfaces. In this method, the actual excitation of an element is represented as the superposition of triangular pulses spaced at the time-step interval used in the simulation, adopting as the size the real surface temperature at the respective time step. Given that the problem is linear, the superposition principle can also be applied and the real responses on each surface can be determined with simple algebraic equations (Equation (A1)). The outcome is the delivery of the same precision as obtained in detailed methods or high-order models with less computing time.

The heat flux on each surface i can be calculated in a given time-step n from the following equation:

$$\dot{q}_{s,i}(n) = \sum_{j=0}^{\infty} f_{1,i}(j)\, T_{s,1}(n-j) + \sum_{j=0}^{\infty} f_{2,i}(j)\, T_{s,2}(n-j) + \sum_{j=0}^{\infty} f_{3,i}(j)\, T_{s,3}(n-j) \qquad \text{(A1)}$$

In Equation (A1) $f_{1,i}$, $f_{2,i}$, and $f_{3,i}$ are the response factors at surface i when the unit triangular pulse excitation is applied at surfaces 1 (top), 2 (bottom), and 3 (pipe), respectively. These response factors represent the averaged heat fluxes for surface i calculated from the surface distribution of the heat flux derived by the transient 2D finite element-based model of ANSYS APDL when unit triangular temperature excitations are applied. To return these surface average values, the node heat fluxes are weighted by the amount of element surface area associated with each node. This operation is internally performed by ANSYS APDL.

As may be inferred from Equation (A1), calculating the heat flux in a given time step calls for a large number of response factors, since the summation must be carried back in time for a period long enough to ensure accurate results. Nonetheless, after a certain number k of response factors, the ratio between the response factor on the surface i when excitation is applied on surface × in a given time step, Ci(k) = fx,i(k)/fx,i(k − 1), and the immediately preceding factoris practically equal: Ci(k) ≈ Ci(k − 1). C is known as the common ratio. It therefore suffices to determine a certain number of response factors to calculate the rest from the common ratio. The summations in Equation (A1) can

consequently be simplified by using the common ratio, as exemplified by Equation (A2) for the top surface (1), where the summation is limited to k terms.

$$
\begin{aligned}
\dot{q}_{s,1}(n) = \\
f_{11}(0)\, T_{s,1}(n) + \sum_{j=1}^{k} T_{s,1}(n-j)[f_{11}(j) - C_1 \cdot f_{11}(j-1)] + C_1 \cdot \dot{q}_{1,1}(n-1) + \\
f_{21}(0)\, T_{s,2}(n) + \sum_{j=1}^{k} T_{s,2}(n-j)[f_{21}(j) - C_1 \cdot f_{21}(j-1)] + C_1 \cdot \dot{q}_{2,1}(n-1) + \\
f_{31}(0)\, T_{s,3}(n) + \sum_{j=1}^{k} T_{s,3}(n-j)[f_{31}(j) - C_1 \cdot f_{31}(j-1)] + C_1 \cdot \dot{q}_{3,1}(n-1)
\end{aligned}
\tag{A2}
$$

To simplify further, the terms referring to time steps prior to the present step can be grouped, yielding:

$$
\dot{q}_{s,1}(n) = f_{11}(0)\, T_{s,1}(n) + f_{21}(0)\, T_{s,2}(n) + f_{31}(0)\, T_{s,3}(n) + h_1
\tag{A3}
$$

where $h_1$ groups all the terms in Equation (A2) referring to time step $n-1$.

Analogously, the heat fluxes across the bottom (2) and pipe (3) surfaces can be calculated as follows:

$$
\dot{q}_{s,2}(n) = f_{12}(0)\, T_{s,1}(n) + f_{22}(0)\, T_{s,2}(n) + f_{32}(0)\, T_{s,3}(n) + h_2
\tag{A4}
$$

$$
\dot{q}_{s,3}(n) = f_{13}(0)\, T_{s,1}(n) + f_{23}(0)\, T_{s,2}(n) + f_{33}(0)\, T_{s,3}(n) + h_3
\tag{A5}
$$

*Appendix A.2. Building the Thermal Model*

The radiant floor model described in the preceding section had to be coupled to a building transient thermal behaviour model to analyse the performance of the floors studied when operating in different dwellings. As noted in Section 2.2, with the model used to describe the building, the equivalent temperature (Teq) and resistance (Req) can be defined as:

$$
T_{eq}(t) = R_{eq} \cdot \left( \frac{T_{s,1}(t)}{R_{cr,1}} + \frac{T_{eq,i,sa}(t)}{R_{eq,i,sa}} + \dot{q}_{cr,i}(t) \right)
\tag{A6}
$$

$$
R_{eq} = \left( \frac{R_{cr,1} \cdot R_{eq,i,sa}}{R_{cr,1} + R_{eq,i,sa}} \right)
\tag{A7}
$$

where:

$$
T_{eq,i,sa}(t) = R_{eq,i,sa} \cdot \left( \frac{T_i(t)}{R_i} + \frac{T_{sa}(t)}{R_e} \right)
\tag{A8}
$$

$$
R_{eq,i,sa} = \left( \frac{R_i \cdot R_e}{R_i + R_e} \right)
\tag{A9}
$$

Applying the energy balance at the capacitance node yields:

$$
C \cdot \frac{dT_{cr,i}(t)}{dt} = \frac{T_{eq}(t) - T_{cr,i}(t)}{R_{eq}}
\tag{A10}
$$

The mathematical procedure for analytically finding the capacitance node temperature in the above differential equation, described by one of the present authors in [60], is expressed in Equation (A11). In this procedure, the excitation is assumed to vary linearly at every simulation time step; this is a reasonable premise inasmuch as excitation data are normally given as discrete values spaced at time-step intervals.

$$
\begin{aligned}
T_{cr,i}(t) = T_{eq}(t) + \tfrac{\tau}{\Delta t} \cdot \left( T_{eq}(t-\Delta t) - T_{eq}(t) \right) + \\
\left[ T_{cr,i}(t-\Delta t) - \left[ T_{eq}(t-\Delta t) + \tfrac{\tau}{\Delta t} \cdot \left( T_{eq}(t-\Delta t) - T_{eq}(t) \right) \right] \right] \cdot e^{-\frac{\Delta t}{\tau}}
\end{aligned}
\tag{A11}
$$

In Equation (A11), $\tau$ = C·Req. If Equation (A6) is substituted into Equation (A11) and the terms are regrouped, Tcr,i can be expressed as a function of surface temperature Ts,1 only as:

$$T_{cr,i} = A_1\,T_{s,1} + B_1 \tag{A12}$$

where:

$$A_1 = \frac{R_{eq}}{R_{cr,1}}\left[\left(1 - \frac{\tau}{\Delta t}\right) + \frac{\tau}{\Delta t}e^{-\frac{\Delta t}{\tau}}\right]$$

$$B_1 = \frac{\tau}{\Delta t}(t - \Delta t)T_{eq} + \left(\frac{R_{eq}}{R_{eq,i,sa}}T_{eq,i,sa}(t) + R_{eq}\cdot\dot{q}_{cr,i}(t)\right)\left[\left(1 - \frac{\tau}{\Delta t}\right) + \frac{\tau}{\Delta t}e^{-\frac{\Delta t}{\tau}}\right] + \left[T_{cr,i}(t - \Delta t) - T_{eq}(t - \Delta t)\left(1 + \frac{\tau}{\Delta t}\right)\right]e^{-\frac{\Delta t}{\tau}}$$

## Appendix B. Values of Building Parameters Chosen for Simulations

### Appendix B.1. Heat Capacity (C)

A building's heat capacity is the result of the combination of density, specific heat, building element thickness, and furnishings. In Table A1 in the International and European standard ISO EN 52016-1 [82], five default values for classifying heat capacity from 'very light' to 'very heavy' are established. Only three heat capacity levels were adopted from the standard for this study (Table A1 heat capacity classification from [82]).

**Table A1.** Heat capacity classification from (UNE-EN ISO 52016-1:2017 2017 [82]).

| Class | Effective Heat Capacity [kJ/(m²K)] |
|---|---|
| Very light | 80 |
| Medium | 165 |
| Very heavy | 370 |

### Appendix B.2. Overall Heat Transfer Coefficient (U-Value)

In the analysis of a broad spectrum of dwellings, three insulation levels, low, medium, and high, were applied to all the envelope elements (Table A2). The aim was to explore a wide range of overall indoor–outdoor heat transfer coefficients (U-value) from very demanding to medium and even low, the third to cover the conditions prevailing in most existing buildings, where insulation values are lower than presently required by building energy codes. The reference values were drawn from the guidelines set out in Spain's Technical Building Code [86].

**Table A2.** U-values and solar heat gain coefficients used for the simulations.

| Insulation Level | U-Value [W/(m²K)] | | | |
| | Wall | Roof | Windows | Windows SHGC |
|---|---|---|---|---|
| Low | 0.79 | 0.47 | 5.70 | 0.72 |
| Medium | 0.53 | 0.31 | 2.80 | 0.63 |
| High | 0.30 | 0.16 | 1.60 | 0.49 |

### Appendix B.3. Overall Heat Transfer Coefficient between Indoor Spaces

The values adopted for the overall heat transfer coefficient for heat transfer between indoor spaces were U = 1.56 W/(m²K) for vertical partitions, characterised by fired clay brick enclosures, and U = 0.71 W/(m²K) for horizontal partitions, comprising floating wood

flooring, a layer of mortar, impact insulation, ceramic inter-joist filling, an air chamber, and plasterboard. Both values were drawn from Spanish legislation [80].

### *Appendix B.4. Area of Interior and Exterior Enclosures*

The areas of each building element were defined as in Table A3.

**Table A3.** Area of building elements by dwelling location, in m$^2$.

|  | Corner Dwelling | Interior Dwelling |
| --- | --- | --- |
| Floor | 90.25 | 90.25 |
| Interior vertical partitions to adyacent spaces | 57 | 85.5 |
| Interior horizontal partitions to upper adyacent spaces | 0 | 90.25 |
| Façade wall | 57 | 28.5 |
| Window area * | 8.55/17.1/34.2 | 4.275/8.55/17.1 |
| Roof | 90.25 | 0 |

\* Three values are shown corresponding to 15%, 30% and 80% of glazed area.

### *Appendix B.5. Radiant Floor Surface Convection-Radiation Heat Transfer Coefficients*

According to Table 3.8 in CIBSE [61] and Table E.1 in [87], the surface thermal resistance in indoor flooring with upward heat flux reached 0.10 m$^2$K/W, deemed here to be the standard value for radiant floors. The same sources cite surface thermal resistance for an indoor ceiling with downward heat flux as 0.17 m$^2$K/W. These were the conditions assumed here to prevail, for the ceiling at issue covered the apartment immediately below the one studied.

### *Appendix B.6. Building Model Boundary Conditions*

#### Appendix B.6.1. Sol-Air Temperature

The sol-air temperatures were calculated on the grounds of hours of solar irradiance and the outdoor air and sky temperatures drawn from the climate data sheet for Madrid available in [88], subsequently interpolated for conversion to a 15-min time step.

The mean radiant temperature was calculated assuming a view factor for the façade walls relative to both sky and ground of 0.5, and the ground and air temperatures were assumed to be equal. The view factor adopted for the roof relative to the sky was 1 and, relative to the ground, it was 0.

The opaque surface absorptivity was assumed to be 0.7.

#### Appendix B.6.2. Adjacent Indoor Space Temperatures

The spaces adjacent to the dwelling were assumed to be heated, with the timing the same as in radiant floor system operation, i.e., during the day, from 08:00 to 23:00, the temperature was 20 °C, declining at night to 17 °C, in keeping with Spanish legislation.

#### Appendix B.6.3. Internal Gain and Ventilation Values

In the model, the convection–radiation heat transfer rate, $\dot{Q}_{cr}$, was calculated as the sum of the indoor source (occupancy and lighting) internal gain values, $\dot{Q}_{int}$ (Table A4), the solar gains due to solar irradiance received through the windows, $\dot{Q}_{sol,int}$, and the ventilation, $\dot{Q}_{vent}$, due to heat gain or loss as a result of indoor–outdoor air changes.

**Table A4.** Indoor sources [W/m$^2$].

| Time of Day | 1 | 2 | 3 | 4 | 5 | 6 | 7 | 8 | 9 | 10 | 11 | 12 | 13 | 14 | 15 | 16 | 17 | 18 | 19 | 20 | 21 | 22 | 23 | 24 |
|---|---|---|---|---|---|---|---|---|---|---|---|---|---|---|---|---|---|---|---|---|---|---|---|---|
| Occupancy | 2.15 | 2.15 | 2.15 | 2.15 | 2.15 | 2.15 | 2.15 | 0.54 | 0.54 | 0.54 | 0.54 | 0.54 | 0.54 | 0.54 | 0.54 | 1.08 | 1.08 | 1.08 | 1.08 | 1.08 | 1.08 | 1.08 | 1.08 | 2.15 |
| Illumination | 0.44 | 0.44 | 0.44 | 0.44 | 0.44 | 0.44 | 0.44 | 1.32 | 1.32 | 1.32 | 1.32 | 1.32 | 1.32 | 1.32 | 1.32 | 1.32 | 1.32 | 1.32 | 2.2 | 4.4 | 4.4 | 4.4 | 4.4 | 2.2 |
| Appliances | 0.44 | 0.44 | 0.44 | 0.44 | 0.44 | 0.44 | 0.44 | 1.32 | 1.32 | 1.32 | 1.32 | 1.32 | 1.32 | 1.32 | 1.32 | 1.32 | 1.32 | 1.32 | 2.2 | 4.4 | 4.4 | 4.4 | 4.4 | 2.2 |
| Total | 3.03 | 3.03 | 3.03 | 3.03 | 3.03 | 3.03 | 3.03 | 3.18 | 3.18 | 3.18 | 3.18 | 3.18 | 3.18 | 3.18 | 3.18 | 3.72 | 3.72 | 3.72 | 5.48 | 9.88 | 9.88 | 9.88 | 9.88 | 6.55 |

The solar gains through the windows, $\dot{Q}_{sol,int}$, were calculated as per Equation (A13):

$$\dot{Q}_{sol,int} = A_w \cdot GR \cdot SHGC \tag{A13}$$

The ventilation-mediated outdoor–indoor heat change, $\dot{Q}_{vent}$, was calculated based on the expression for ventilation gains or losses as:

$$\dot{Q}_{vent} = \dot{V} \cdot \rho_{air} \cdot Cp_{air} \cdot (T_{air,e} - T_{cr,i}) \tag{A14}$$

The air flow rate was estimated to ensure a mean $CO_2$ concentration of under 900 ppm, a requirement that can be met with a flow rate of approximately 12.5 L/(s·person) [61]. The total air flow rate for the dwelling given a mean occupancy of three occupants, a value consistent with the indoor sources attributable to occupancy assumed, came to 3.5 L/s or 0.5 ACH.

*Appendix B.7. Radiant Floor Operation. Timing, Set Points, and Water Temperature*

Heating systems may shut down and started up several times during the day, either after reaching a set point temperature or in line with scheduled operating times. The set point temperature established for this study was 20 °C, measured at the node representing the indoor space in the dwelling (Tcr,i).

The operating times were defined, based on Spanish legislation [89], as 08:00 to 23:00; outside of this window, the heating was assumed to be off.

During radiant floor operation, the mean water temperature in the pipes was assumed to be 40 °C, whereas, when it was off, the water temperature varied freely.

**Appendix C. Explanation of Start-Up Lag Time Discontinuity**

The discontinuities in the start-up lag times with rising thermal resistance in the poorest-performing dwellings represented on the curves in Figure 16 are explained graphically in Figure A1. In this figure, the operative temperature is plotted against the lag time for two coverings in one of the poorest-performing dwellings at the point at which one of the discontinuities was observed. The sole (and small) difference between the two cases was the thermal resistance of the covers, 0.114 m$^2$K/W in one and 0.117 m$^2$K/W in the other. Although the operative temperatures were practically identical in the two, enlarging the area of the discontinuity revealed that when the flooring resistance was 0.114 m$^2$K/W, the operative temperature was slightly higher and reached 20 °C within the 41 h limit, whereas, when the resistance was 0.117 m$^2$K/W, the operative temperature reached after 141 h was very close, but not equal, to 20 °C. This took a further 24 h. This behaviour was observed in the poorest-performing dwellings only, where the indoor operative temperatures declined sharply overnight due to outward heat loss.

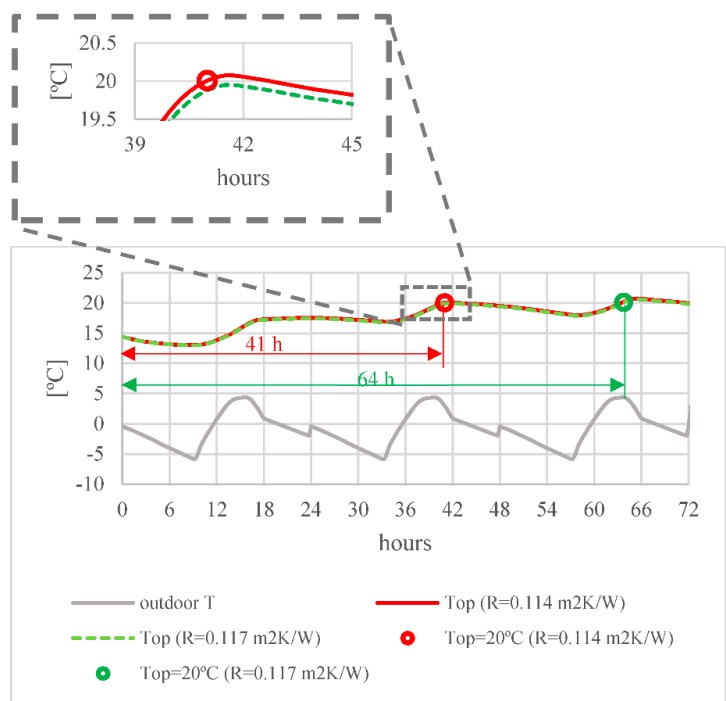

**Figure A1.** Explanation of start-up lag time discontinuity (variation in operative temperature in one of the poorest-performing dwellings with two coverings, differing in thermal resistance: 0.114 m$^2$W/K and 0.117 m$^2$W/K.

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
