# Peer review of "Influence of Wood Properties and Building Construction on Energy Demand, Thermal Comfort and Start-Up Lag Time of Radiant Floor Heating Systems"

_applsci, doi:10.3390/app12052335_

Round 1

Reviewer 1 Report

The article is well constructed and impressive. It can only be interpreted that the average difference in transition time between wood and granite cladding is 3 in 75% of the residences (page 17 and conclusion). Can a certain critical limit be mentioned? An explanation would be helpful.

Author Response

Reviewer 1

The article is well constructed and impressive. It can only be interpreted that the average difference in transition time between wood and granite cladding is 3 in 75% of the residences (page 17 and conclusion). Can a certain critical limit be mentioned? An explanation would be helpful.

Response: We are very grateful to the reviewer 1’s useful and interesting suggestion.

We have added a paragraph on page 17 explaining in a little more detail the results and why it is difficult to establish a critical limit as this may depend on the particular conditions of each flat. We have also added a comment in the conclusions, explaining the above a little more concisely

Reviewer 2 Report

The Paper presents a very complex study on the influence of wood properties. The paper provides useful data for researchers as well as practitioners. 

The method and research approach is clearly justified.  Data analysis covers almost all possible aspects and is sufficiently detailed in order to explain results. 

However, it is an extremely long paper. Probably some basic theory and literature review could be optimized. 

Author Response

Reviewer 2

The Paper presents a very complex study on the influence of wood properties. The paper provides useful data for researchers as well as practitioners. 

The method and research approach is clearly justified.  Data analysis covers almost all possible aspects and is sufficiently detailed in order to explain results. 

However, it is an extremely long paper. Probably some basic theory and literature review could be optimized.

Response: We very much appreciate the helpful and constructive comments of reviewer 2.

In order to reduce the length of the article, we have deleted an entire paragraph from the introduction in which a brief review of the historical evolution of the main aspects on which research on radiant floors has focused was made. We believe that its elimination does not alter the most relevant aspects of the article and it does shorten the reading of it a bit. In this way, the focus of this research can be found more quickly.

Reviewer 3 Report

Dear Authors,

Very interesting approach tackling the current issues in wood radiant floors application and their thermal performance.

Even though the radiant floor heating is popular in cold climates, it is widely used in the temperate climates also, especially nowadays with combination of geothermal heat pumps. That’s the reason that the presented analysis is considered of great importance. The analysis covers the different cases or scenarios and assessed the impact of different wooden material coverings used for radiant floors in term of thermal performance. And also compared with material with high thermal conductivity such as: granite, insofar, in terms of: energy demand, thermal comfort and start-up lag time (which is considered as very innovative approach to the problem). Such complex analysis is very challenging and the authors have covered very detail the presented model. Especially the simulation model was quite clear and easy to understand. The approach of used transient thermal model was very innovative. The authors have also provided data for experimental validation of the model (as well as the RMS error calculated between the model values and experimental values for each water inflow temperature).

I would like that authors emphases the strengths and weaknesses of applied approach, that need to be highlighted. It is not obligatory but would contribute the overall significance of presented analysis.

Author Response

Reviewer 3

Very interesting approach tackling the current issues in wood radiant floors application and their thermal performance.

Even though the radiant floor heating is popular in cold climates, it is widely used in the temperate climates also, especially nowadays with combination of geothermal heat pumps. That’s the reason that the presented analysis is considered of great importance. The analysis covers the different cases or scenarios and assessed the impact of different wooden material coverings used for radiant floors in term of thermal performance. And also compared with material with high thermal conductivity such as: granite, insofar, in terms of: energy demand, thermal comfort and start-up lag time (which is considered as very innovative approach to the problem). Such complex analysis is very challenging and the authors have covered very detail the presented model. Especially the simulation model was quite clear and easy to understand. The approach of used transient thermal model was very innovative. The authors have also provided data for experimental validation of the model (as well as the RMS error calculated between the model values and experimental values for each water inflow temperature).

I would like that authors emphases the strengths and weaknesses of applied approach, that need to be highlighted. It is not obligatory but would contribute the overall significance of presented analysis.

Response: We are very thankful for the interesting and helpful comments of reviewer 3.

At the end of the conclusions we have added a paragraph in which we express what we consider to be the most relevant strengths of the methodology used and make a very brief discussion about other possible approaches to the study of radiant floors. Also we explain why we believe that the best way to approach a general study, such as the one presented in this article, is the one we have taken.